# Photoreceptor avascular privilege is shielded by soluble VEGF receptor-1

Ling Luo[1,2]*, Hironori Uehara[1], Xiaohui Zhang[1], Subrata K Das[1], Thomas Olsen[1], Derick Holt[1], Jacquelyn M Simonis[1], Kyle Jackman[1], Nirbhai Singh[1], Tadashi R Miya[1], Wei Huang[1], Faisal Ahmed[1], Ana Bastos-Carvalho[3], Yun Zheng Le[4], Christina Mamalis[1], Vince A Chiodo[5], William W Hauswirth[5], Judit Baffi[3], Pedro M Lacal[6], Angela Orecchia[7], Napoleone Ferrara[8†], Guangping Gao[9,10], Kim Young-hee[3], Yingbin Fu[1], Leah Owen[1], Romulo Albuquerque[3], Wolfgang Baehr[1], Kirk Thomas[10], Dean Y Li[10], Kakarla V Chalam[11], Masabumi Shibuya[12], Salvatore Grisanti[13], David J Wilson[14], Jayakrishna Ambati[3], Balamurali K Ambati[1]*

[1]Moran Eye Center, University of Utah, Salt Lake City, United States; [2]Department of Ophthalmology, The 306th Hospital of PLA, Beijing, China; [3]Ophthalmology and Visual Sciences, University of Kentucky, Lexington, United States; [4]Department of Medicine and Harold Hamm Oklahoma Diabetes Center, University of Oklahoma Health Sciences Center, Oklahoma City, United States; [5]Ophthalmology, University of Florida-Gainesville, Gainesville, United States; [6]Laboratory of Molecular Oncology, IDI-IRCCS, Istituto Dermopatico dell'Immacolata-IRCCS, Rome, Italy; [7]Laboratory of Molecular and Cell Biology, Istituto Dermopatico dell'Immacolata-IRCCS, Rome, Italy; [8]Pharmaceutical Research, Genentech, South San Francisco, United States; [9]Gene Therapy Center, University of Massachusetts Medical School, Worcester, United States; [10]Department of Internal Medicine, Division of Cardiology, University of Utah, Salt Lake City, United States; [11]Department of Ophthalmology, University of Florida, Jacksonville, Jacksonville, United States; [12]Molecular Oncology, Institute of Physiology and Medicine, Jobu University, Takasaki, Japan; [13]Ophthalmology, University of Luebeck, Luebeck, Germany; [14]Department of Ophthalmology, Casey Eye Institute, Oregon Health & Science University, Portland, United States

*For correspondence: ling. luoling1208@gmail.com (LL); bambati@gmail.com (BKA)

†Present address: Department of Pathology, University of California, San Diego, La Jolla, United States

Competing interests: The authors declare that no competing interests exist.

## Abstract

Optimal phototransduction requires separation of the avascular photoreceptor layer from the adjacent vascularized inner retina and choroid. Breakdown of peri-photoreceptor vascular demarcation leads to retinal angiomatous proliferation or choroidal neovascularization, two variants of vascular invasion of the photoreceptor layer in age-related macular degeneration (AMD), the leading cause of irreversible blindness in industrialized nations. Here we show that sFLT-1, an endogenous inhibitor of vascular endothelial growth factor A (VEGF-A), is synthesized by photoreceptors and retinal pigment epithelium (RPE), and is decreased in human AMD. Suppression of sFLT-1 by antibodies, adeno-associated virus-mediated RNA interference, or Cre/lox-mediated gene ablation either in the photoreceptor layer or RPE frees VEGF-A and abolishes photoreceptor avascularity. These findings help explain the vascular zoning of the retina, which is critical for vision, and advance two transgenic murine models of AMD with spontaneous vascular invasion early in life.

**eLife digest** The inner surface of the vertebrate eye is lined with a multilayered structure known as the retina. The bottom layer of the retina is composed of rods and cones—neurons that are directly sensitive to light—and is called the photoreceptor layer. Rods function primarily in dim light and provide black-and-white vision, while cones support daytime vision and are responsible for colour perception. Unlike the upper layers of the retina, the photoreceptor layer does not contain blood vessels: oxygen and nutrients are instead provided by a structure just underneath the retina called the choroid.

The eye relies on the rods and cones converting light into electrical signals, and the photoreceptor layer must remain free of blood vessels for this process to work properly. If blood vessels extend into the photoreceptor layer from rest of the retina (which is above it) or the choroid (below), they can disrupt the retina and give rise to a condition called age-related macular degeneration, which is a leading cause of irreversible blindness in adults.

Within the eye, the development of new blood vessels from pre-existing vessels is stimulated by a protein known as vascular endothelial growth factor A (VEGF-A), while an inhibitor protein called sFLT-1 prevents the growth of new blood vessels in the other tissues of the eye like the cornea. However, it has not been clear what keeps the photoreceptor layer (and also the cells that support the photoreceptor layer) free of blood vessels, and what happens to disrupt this process of vascular demarcation in age-related macular degeneration.

Now, Luo et al. reveal that cells in the photoreceptor layer produce sFLT-1, and that the levels of this protein are indeed reduced in people with age-related macular degeneration. Using genetic and pharmacological methods, they show that a reduction in sFLT-1 triggers blood vessels to grow into the photoreceptor layer from above or below. Luo et al. also report two new genetic mouse models in which blood vessels form spontaneously in the photoreceptor layer at an early age, which should prove useful for further research into age-related macular degeneration.

## Introduction

Vascular demarcations in the eye are both striking and vital for vision. The completely avascular photoreceptor layer of the retina lies between the vascular inner retina (on its apical aspect) and the highly vascularized and permeable fenestrated choriocapillaris (*Blaauwgeers et al., 1999*) (on its basal aspect) (*Figure 1—figure supplement 1*). The choroid, which has the highest flow of any vascular bed (*Bill et al., 1983*), fulfills the metabolic and oxygen demands of the retinal pigmental epithelium (RPE) cells and photoreceptors. In age-related macular degeneration (AMD), angiogenic vessels from the inner retina extend into the photoreceptors (retinal angiomatous proliferation, or RAP), or from choroidal vessels breaching through the complex of the RPE and Bruch's membrane (BrM) into the subretinal space (choroidal neovascularization, CNV). Either can disrupt retinal structure and function, causing irreversible vision loss (*Ambati et al., 2003b*). Although AMD is the leading cause of irreversible blindness in the West (*Ambati et al., 2003b*), its etiology has not been fully elucidated.

Understanding the molecular underpinnings of the physiological vascular zoning ability of the retina has been a challenge for retinal biology for decades and may provide a better understanding of AMD pathogenesis. Vascular endothelial growth factor A (VEGF-A) is a potent angiogenic factor with a crucial role in both normal and pathologic vascular growth within the eye (*Saint-Geniez et al., 2008*; *Saint-Geniez et al., 2009*; *Rajappa et al., 2010*; *Carmeliet Jain, 2011*; *Potente et al., 2011*), and its increased expression is linked to pathologic proliferation of abnormal, highly permeable vessels into the avascular photoreceptor layers (*Amin et al., 1994*; *Ambati et al., 2003b*; *Rajappa et al., 2010*). A soluble isoform of VEGF receptor-1 (sVEGFR-1, also known as sFLT-1) acts as a naturally occurring inhibitor of VEGF-A-mediated angiogenesis (*Kendall and Thomas, 1993*). sFLT-1 negatively regulates VEGF-A action: first, it binds and sequesters VEGF-A; and second, by heterodimerizing with membrane-bound VEGF-receptor-2, it prevents VEGF-A occupancy and subsequent signal transduction (*Kendall et al., 1996*). Excess levels of sFLT-1 strongly inhibit CNV in mice and monkeys (*Lai et al., 2005*; *Lukason et al., 2011*). In addition, we previously demonstrated that endogenous sFLT-1 is both necessary and sufficient for corneal avascularity (*Ambati et al., 2006*). Therefore, we hypothesized that avascular

photoreceptor privilege depends on sFLT-1 and that a deficiency of endogenous sFLT-1 might be involved in CNV or RAP development.

## Results

### sFLT-1 is decreased in AMD RPE and photoreceptors

Although the specificity of soluble FLT-1 antibody had been verified, we further confirmed this by ELISA as it is crucial to distinguish it from membrane FLT-1 (*Figure 1—figure supplement 2*). We then compared sFLT-1 expression in healthy human eyes and eyes with AMD or RAP (patient information is shown in *Table 1*) by two different immunostaining methods. Both results demonstrated that soluble FLT-1 was strikingly decreased in the RPE and photoreceptor layers in all AMD (n = 3) and RAP eyes (n = 3) compared with age matched controls, respectively (*Figure 1A*, *Figure 2—figure supplement 3*); conversely, as has been previously reported, VEGF-A was upregulated significantly in the retina of AMD eyes (*Ambati et al., 2003b*).

### Photoreceptors and RPE synthesize and express sFLT-1

We next assessed sFLT-1 expression in the retina. In situ hybridization and immunostaining confirmed the presence of sFLT-1 mRNA and protein, respectively, in photoreceptors and RPE, which indicated that both cell types can synthesize and express sFLT-1 (*Figure 1B,C*). Compared with VEGF-A (*Figure 1C*), the relative expression pattern of soluble FLT-1 to VEGF is higher in the photoreceptors than in the inner vascular layers of the retina. A theoretical implication is that VEGF is prominent in the inner vascularized retina, the layer which harbors blood vessels and neurons, while sFLT-1 is prominent in the outer avascular retina, consistent with our initial hypothesis. Furthermore, sFLT-1 was principally detected on the basal aspect of the RPE–BrM complex, facing the vascularized choroid (*Figure 1C,D*). In contrast, in the RPE, VEGF-A localizes on both basal and apical surfaces, consistent with a prior report (*Blaauwgeers et al., 1999*). This polarized distribution is reminiscent of sFLT-1 expression in the cornea, where the highest levels of sFLT-1 are found in the perilimbal region counterposing the vascularized conjunctiva (*Ambati et al., 2007*). This is consistent with VEGF's vasculotrophic role in the choroid and neurotrophic role in the photoreceptors (*Saint-Geniez et al., 2008*). The different ratio and polarities of VEGF-A and sFLT-1 would likely strike a reasonable balance to maintain a healthy photoreceptor layer and choriocapillaris while preventing vascular invasion of the subretinal space.

### Anti-FLT-1 antibodies induce CNV

Given the localization and expression patterns of sFLT-1, we sought to determine if suppression of subretinal sFLT-1 in mice would induce subretinal angiogenesis or CNV. First, we injected a neutralizing antibody against FLT-1 into the subretinal space in wild type mice. This resulted in increased free VEGF-A levels (n = 10; p<0.005) and induced CNV (n = 10, p<0.00005; *Figure 2A,B*), suggesting that sequestration of VEGF-A by FLT-1 maintains subretinal avascularity. Next, we administered FLT-1 neutralizing antibody (which occupies domains 1–4 of this receptor) via subretinal injection in *Flt-1 tk⁻/⁻* mice, which are deficient in the tyrosine kinase domain of membrane-bound FLT-1 (*Hiratsuka et al., 1998*). The results demonstrated that the induced CNV in *Flt-1 tk⁻/⁻* mice was accompanied with higher VEGF levels (n = 10; p<0.005; *Figure 2C,D*), strongly suggesting that CNV occurs due to disruption of sFLT-1, but not of mFLT-1 activity (which is inactive in these mice). This accords with prior studies which have found that soluble but not membrane-bound FLT-1 plays a key role in anti-angiogenic regulation, based on the finding that germline *Flt-1* deletion leads to embryonic lethality due to vascular endothelial hyperplasia, while membrane-bound *Flt-1* lacking the tyrosine kinase domain is sufficient for normal vascular development in mice (*Fong et al., 1995*; *Kappas et al., 2008*).

**Table 1.** Demographic information for human globes

| Normal | Normal | Normal | Normal | Normal | AMD (CNV) | AMD (dry) | AMD | RAP | RAP | RAP |
|---|---|---|---|---|---|---|---|---|---|---|
| 25/M | 71/M | 79/F | 80/F | 89/F | 80/F | 86/F | 88/F | 67/M | 86/F | 86/F |

All humans are Caucasians.

AMD: age-related macular degeneration; CNV: choroidal neovascularization; F: female; M: male; RAP: retinal angiomatous proliferation.

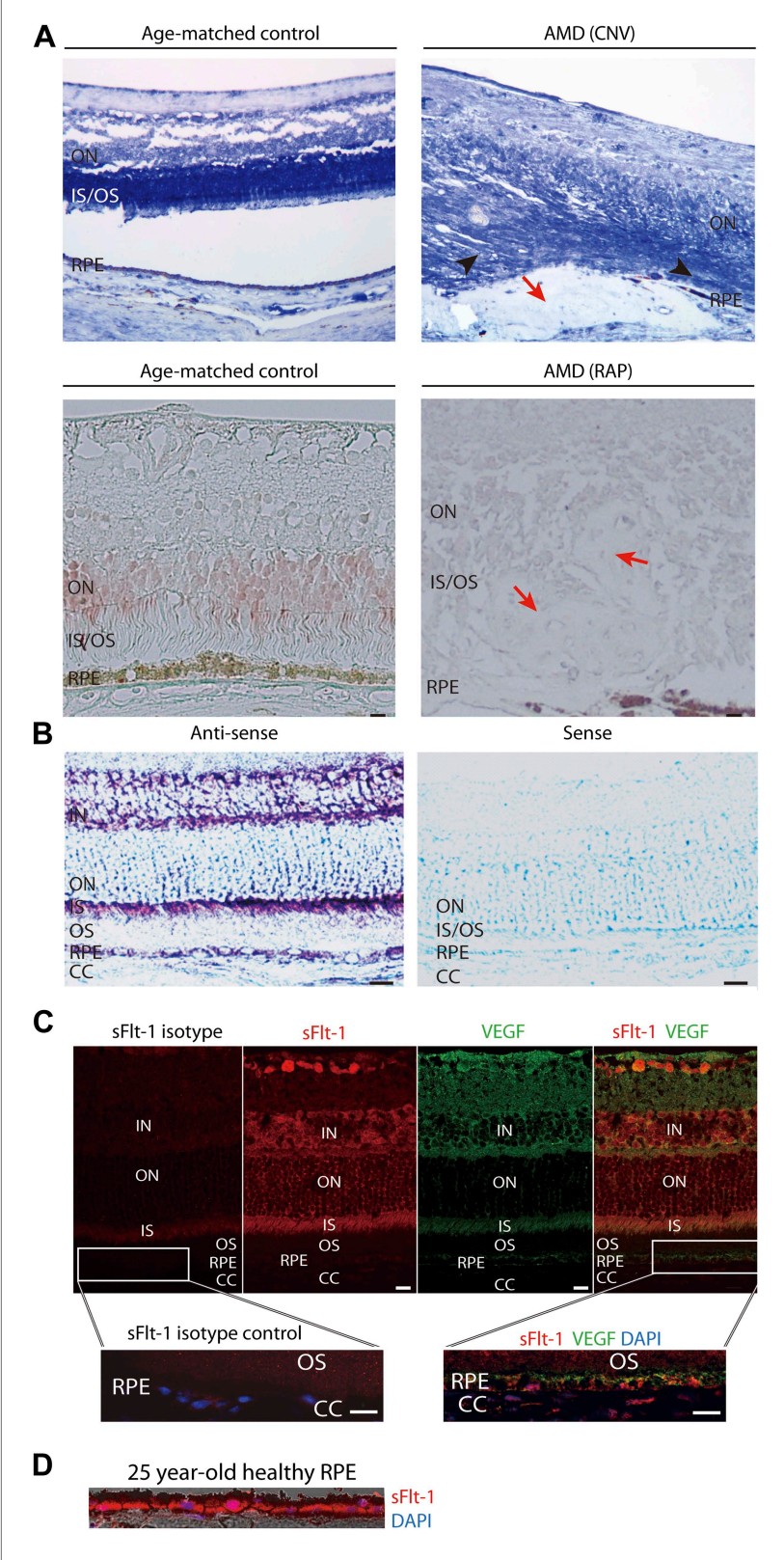

**Figure 1**. Soluble FLT-1 expression is reduced in the photoreceptors and RPE from AMD eyes. (**A**) Representative immunostaining images show soluble FLT-1 expression was significantly decreased in an age-related macular

*Figure 1. Continued on next page*

*Figure 1. Continued*

degeneration (AMD) eye with choroidal neovascularization (CNV) (top, 80 years old, female) and retinal angiomatous proliferation (RAP) (bottom, 67 years old, female) compared with aged-matched controls (80 years old and 71 years old, respectively, females; black arrow heads point to the photoreceptors; red arrows point to the CNV or RAP lesions). (**B**) In situ hybridizations show the localization of *sFlt-1* (purple/blue) in the photoreceptors and retinal pigment epithelium (RPE) layers in Balb/c mice. (**C**) Immunohistochemistry (IHC) staining shows sFLT-1 and VEGF expression in wild type mice. Higher relative expression of sFLT-1 to VEGF is observed in the photoreceptors. The magnified images (bottom) from the framed area showed that soluble FLT-1 is expressed in the basal side of the RPE layer. (**D**) Representative IHC image shows sFLT-1 is expressed in the basal side of the RPE layer in a young adult healthy human eye (25 years old, male). CC: choriocapillaris; IN: inner nuclear layer; IS: inner segment layer; ON: outer nuclear layer; OS: outer segment layer. Arrows point to the RPE layer. Scale bar: 10 µm.

The following figure supplements are available for figure 1:

**Figure supplement 1**. Avascularity of the outer retina (photoreceptors and RPE and BrM) surrounded by the inner retina with abundant vessels and the highly vascularized choroid in a normal human eye.

**Figure supplement 2**. sFLT-1 antibody specifically binds to the unique motif of sFLT-1.

**Figure supplement 3**. sFLT-1 expression is significantly decreased in RPE from an AMD eye with CNV (88 years old, female) compared with the age-matched control (89 years old, female).

**Figure supplement 4**. H&E staining images show the histology of two human RAP eyes (arrows point to the RAP lesion).

In addition to VEGF-A, sFLT-1 also binds VEGF-B and placental growth factor (PlGF) (*Malik et al., 2006*; *Fischer et al., 2008*). To determine if PlGF is essential for CNV, we performed subretinal injections of anti-FLT-1 and isotype control antibody in wild type mice and found no significant difference in PlGF expression between both groups at day 3 (n = 9; *Figure 2E*) or at day 10 (undetectable levels). Taken together with the above finding that FLT-1 antibody induced CNV in *Flt-1 tk*$^{-/-}$ mice (in which VEGF-B signaling would be expected to be inoperative), this indicates that CNV induced by FLT-1 blockade is mediated by desequestration of VEGF-A, not of PlGF or VEGF-B. Moreover, neither PlGF nor VEGF-B are able to compensate for VEGF-A during its blockade, and mice lacking either factor display only minor developmental defects (*Malik et al., 2006*; *Zhang et al., 2009*). Furthermore, it is well established that, in contrast to VEGF-A, VEGF-B is neither induced by hypoxia nor essential to angiogenesis (*Hoeben et al., 2004*; *Zhang et al., 2009*). Although VEGF-B is dispensable for blood vessel growth, it is critical for blood vessel survival in pathological conditions (*Zhang et al., 2009*). However, our findings do not exclude a role for VEGF-B in RAP or CNV. Knocking down sFLT-1 could increase VEGF-B activity and promote the longevity of CNV lesions after formation rather than promote angiogenesis. However, this pro-survival function is mediated through membrane-bound FLT-1.

Considering the high affinity sFLT-1 has for VEGF, it is crucial to verify that blocking antibodies can displace VEGF from bound sFLT-1 and release free VEGF. Moreover, anti-VEGF antibodies may affect the quantification of VEGF and PlGF, and another consideration is that the particular ELISA used may measure 'non-free' VEGF or PlGF (bound to sFLT-1) as well. To clarify these questions, we performed the following series of experiments. First, we determined whether the neutralizing anti-FLT-l antibody affected the measurement of mouse VEGF-A and PlGF-2, the only isoform of this protein present in mice by ELISA (*Figure 2—figure supplement 1*). We did not find any significant difference.

Next, to determine whether our ELISA would detect VEGF that is bound to FLT-1, we examined the effect of excess recombinant FLT-1 protein on the detection of VEGF-A and PlGF-2 by ELISA (*Figure 2—figure supplement 2*). In this assay, almost all VEGF-A (62.5 pg/ml) would be expected to bind FLT-1 (100 ng/ml) based on an assumed Kd = 10 pM (free VEGF-A can be estimated to be 1.36 pg/ml by Michaelis–Menten kinetics). The ELISA showed less detection of VEGF-A and PlGF after saturation with excess recombinant FLT-1, that is, it did not detect non-free VEGF-A and non-free PlGF-2. These data demonstrate that non-free VEGF or non-free PlGF-2 is not being detected at significant levels by our ELISA technique.

Finally, we determined whether anti-FLT-1 neutralizing antibody released VEGF-A but not PlGF-2 from recombinant FLT-1 (*Figure 2—figure supplement 3*). After coating ELISA plates with FLT-1 and

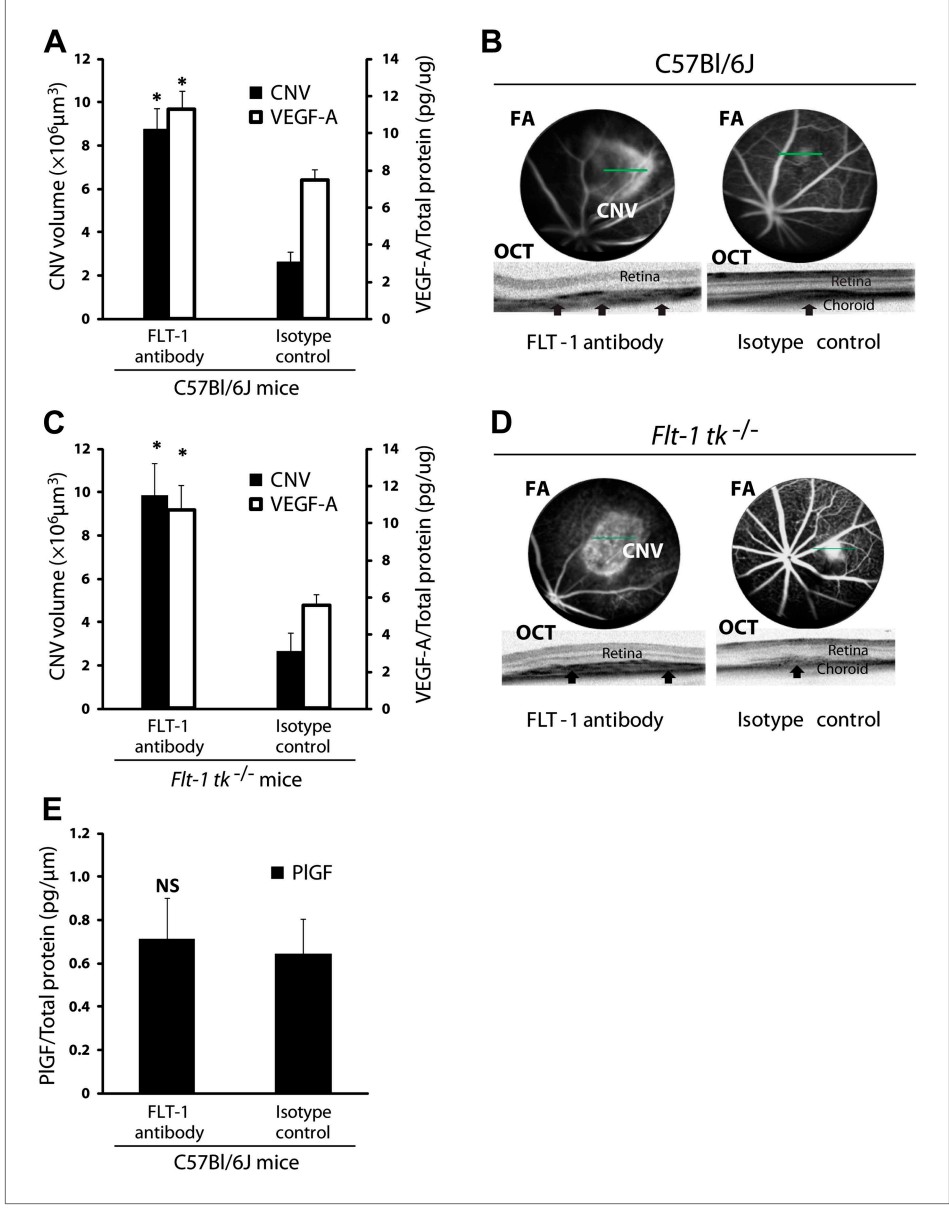

**Figure 2**. Suppression of sFLT-1 by neutralizing antibodies induces CNV while elevating levels of VEGF. VEGF-A levels were tested by ELISA of retinal pigment epithelium (RPE)/choroid tissue lysates harvested immediately after CNV images were taken in vivo. (**A** and **C**) Subretinal injections of FLT-1 antibody induced CNV and increased VEGF-A levels in both wild type and *Flt-1 tk*[-/-] mice. (**B** and **D**) Representative FA (at a phase of 3 min after fluorescein injection) and OCT images. Arrows point to CNV. (**E**) ELISA for PlGF showed no significant difference 3 days after subretinal injections of FLT-1 neutralizing antibody and isotype control. CNV: choroidal neovascularization; FA: fluorescein angiography; OCT: optical coherence tomography.

The following figure supplements are available for figure 2:

**Figure supplement 1**. Goat anti-mouse FLT-1 neutralizing antibody does not affect ELISA assessment of mouse VEGF-A and mouse PlGF-2.

**Figure supplement 2**. Excess recombinant FLT-1 inhibits VEGF-A and PlGF-2 detection.

**Figure supplement 3**. FLT-1 neutralizing antibody incubation resulted in release of VEGF-A from recombinant FLT-1 in vitro.

then incubating with VEGF-A or PlGF-2, Flt-1 neutralizing antibody was added. The results demonstrate that FLT-1 neutralizing antibody released VEGF-A but not PlGF-2 from recombinant FLT-1. Although the exact reason for this is unknown, it is possible that FLT-1 neutralizing antibody may change the FLT-1 conformation, affecting binding of FLT-1 to VEGF-A but not PlGF-2.

## AAV.shRNA.*sFlt-1* induces CNV

Our second strategy was a post-transcriptional selective knockdown of sFLT-1 via subretinal injection of adeno-associated viral (AAV) delivery of a short hairpin RNA (shRNA) targeting sFLT-1 (AAV.shRNA.*sFlt-1.Gfp*). We have previously shown that this shRNA is selective to sFLT-1 and does not affect membrane-bound FLT-1 (*Ambati et al., 2006*). We further confirmed this by western blot (*Figure 3—figure supplement 1*). As controls, shRNA (AAV.shRNA.*Gfp*), viral vector expressing non-specific cargo (AAV.*Gfp*), or phosphate buffered saline (PBS) was injected into the subretinal space. Subretinal delivery of AAV.shRNA.*sFlt-1.Gfp* demonstrated efficient transduction of the RPE and the photoreceptor layer (*Figure 3—figure supplement 2*). We analyzed CNV and measured VEGF-A at 4 weeks after subretinal injection instead of 10 days as above, as AAV-mediated subretinal gene delivery usually takes a longer time before expression (*Shao et al., 2012*). Compared to controls, treatment with AAV.shRNA.*sFlt-1.Gfp* resulted in downregulation of sFLT-1 (*Figure 3A*), increased free VEGF-A (n = 10; p<0.05; *Figure 3B*), and induced CNV (n = 10; p<0.05; *Figure 3B,C*, *Figure 3—figure supplement 3*). Furthermore, as we reported recently (*Luo et al., 2013*), AAV.shRNA.*sFlt-1* induced primary CNV (defined as a CNV that initially occurred at the injection site), as well as secondary CNV (defined as a lesion that is more than ½ a disc diameter from the primary CNV lesion, and that shows no leakage on fluorescein angiography (FA) at least 2 weeks after injection) 4 weeks after injection. This phenomenon was not found in the controls, although insignificant CNV developed initially due to injury. The length of time required for secondary CNV occurrence might derive from the time to and long duration of AAV-mediated gene expression in the mouse eye which has previously been demonstrated (*Schlichtenbrede et al., 2003*; *Kjellstrom et al., 2007*; *Shao et al., 2012*). To further determine if CNV induced by sFLT-1 inhibition is attributable to VEGF-A, we repeated the above set of AAV-mediated sFLT-1 knockdown studies in *Vegfa*<sup>lox/lox</sup> mice (*Gerber et al., 1999*). The reduction of VEGF protein in RPE cells via subretinal delivery of a plasmid encoding Cre recombinase (pCre) in *Vegfa*<sup>loxp/loxp</sup> mice was demonstrated by immunohistochemistry (IHC) staining (*Figure 3—figure supplement 4*). Cre expression was verified by western blot (*Figure 3—figure supplement 5*). Increased VEGF-A expression in the RPE/choroid (n = 10; p<0.0005) and CNV (n = 10; p<0.05) in *Vegfa*<sup>lox/lox</sup> mice treated with subretinal AAV.shRNA.*sFlt-1* (*Figure 3D*) was suppressed by subretinal delivery of pCre. This confirms that the release of sequestered VEGF-A is requisite for CNV development following sFLT-1 antagonism. We observed that the sFLT-1 fluorescence signal became stronger after subretinal injection in treatment or control (*Figure 3A*) eyes compared to normal eyes without injury (*Figure 1C*). The higher sFLT-1 expression might be due to the inflammatory reaction following injury.

Treatment with siRNAs can affect several cytokines, such as interferon-gamma (IFN-γ) and interleukin 12 (IL-12), via activation of toll-like receptor 3 (TLR3), and suppress CNV as a general siRNA class effect (*Kleinman et al., 2008*). We found that AAV.shRNA, targeting either *sFlt-1* or a non-specific sequence, up-regulated IFN-γ (n = 10; p<0.01) and IL-12 (n = 10; p<0.001; *Figure 3E* and *Figure 3—figure supplement 6*), both of which are induced by TLR3. However, non-specific AAV.shRNA.*Gfp* failed to induce CNV in wild type mice (n = 10; *Figure 3B*), suggesting that CNV formation in AAV.shRNA.*sFlt-1.Gfp* treated mice is due not to non-specific immune activation but rather to sFLT-1 suppression. To confirm this, we repeated the described experiments in Tlr3-knockout mice (*Tlr3*<sup>–/–</sup> mice). Subretinal injection of AAV.shRNA.*sFlt-1.Gfp* (n = 10) but not non-specific AAV.shRNA.*Gfp* (n = 10) increased VEGF-A levels (p<0.05) and induced CNV (p<0.005) in *Tlr3*<sup>–/–</sup> mice (*Figure 3F*). Levels of IFN-γ and IL-12 were not significantly different in either group (n = 10; *Figure 3E* and *Figure 3—figure supplement 6*), suggesting that CNV is due to sFLT-1 knockdown rather than non-specific immune activation.

## Selective Cre/*lox* FLT-1 ablation induces CNV and RAP

The third strategy was genomic deletion. We suppressed sFLT-1 by a Cre/*lox*-mediated conditional gene ablation because *Flt-1* deletion is lethal (*Fong et al., 1995*). Subretinal injection of pCre in *Flt-1*<sup>lox/lox</sup> mice reduced sFLT-1, induced CNV (n = 10; p<0.01), and increased free VEGF-A in the RPE/choroid (n = 8; p<0.01) (*Figure 4A*).

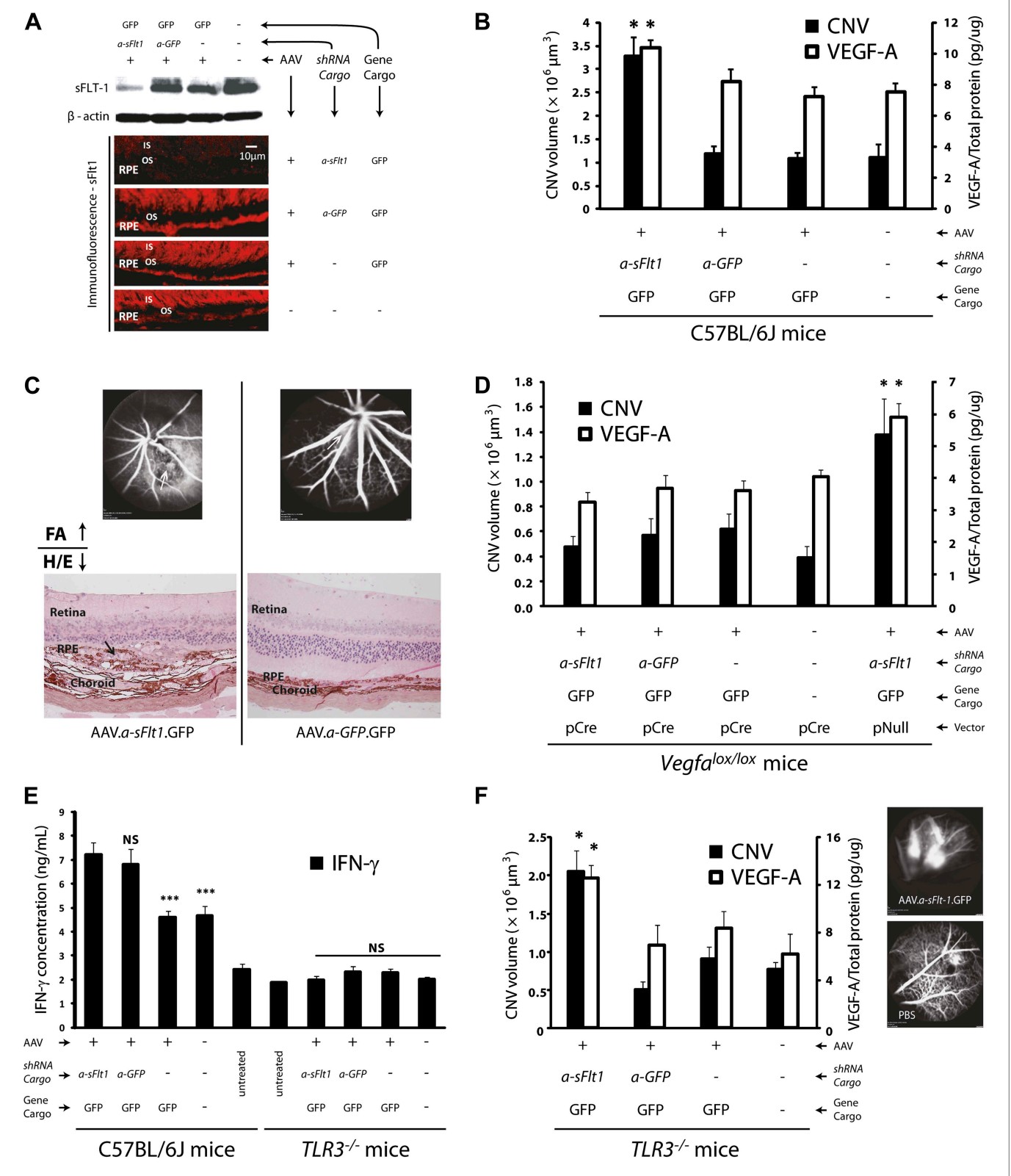

**Figure 3**. Subretinal adeno-associated viral delivery of short hairpin RNA targeting sFLT-1 induces VEGF-A and CNV. VEGF-A levels from retinal pigment epithelium (RPE)/choroid tissues were tested by ELISA from tissue lysates harvested immediately after choroidal neovascularization (CNV) images were

*Figure 3. Continued on next page*

*Figure 3. Continued*

taken in vivo. (**A**) Western blot (top) and immunohistochemistry (IHC) staining of sFLT-1 results confirmed that sFLT-1 was knocked down by AAV.shRNA.*sFlt-1.Gfp* but not in controls at 4 weeks. (**B**) Subretinal AAV.shRNA.*sFlt-1.Gfp* induced CNV and VEGF-A compared to controls in wild type mice (4 weeks). CNV was observed in controls due to disruption of Bruch's membrane by subretinal injection. (**C**) Representative fluorescein angiography (FA) and H&E staining images (40×) (arrows point to leakage or neovessels) show CNV is more extensive after AAV.shRNA.*sFlt-1.Gfp* treatment compared to controls. (**D**) CNV was not induced by subretinal AAV.shRNA.*sFlt-1* when VEGF-A release was suppressed (4 weeks) by pCre in *Vegfa$^{lox/lox}$* mice compared with pNull control in which VEGF levels rose and CNV was induced. (**E**) IFN-γ levels increased in AAV.shRNA.*sFlt-1.Gfp* and AAV.shRNA.*Gfp* (4 weeks) compared with AAV.*Gfp* and phosphate buffered saline (PBS) in C57BL/6J mice. No significant difference was found among groups in the *Tlr3$^{-/-}$* mice (4 weeks). (**F**) Subretinal AAV.shRNA.*sFlt-1.Gfp* induced VEGF-A and CNV (4 weeks) in *Tlr3$^{-/-}$* mice. Representative FA images show the leakage in each group. VEGF-A levels were tested by ELISA using RPE/choroid complex samples. *p<0.05; **p<0.005; ***p<0.0005. NS: no significant difference. Scale bar: 10 µm.

The following figure supplements are available for figure 3:

**Figure supplement 1**. Western blot shows that AAV.shRNA.*sFlt-1.Gfp* specifically affects sFLT-1 but not mFLT-1.

**Figure supplement 2**. AAV.shRNA.*sFlt-1.Gfp* (A) transfected more than 3 out of 4 quadrants and targeted the photoreceptors, RPE, and BrM by subretinal injection (2 weeks). (B) Sham control.

**Figure supplement 3**. Representative images show different sizes of CNV in AAV.shRNA.*sFlt-1.Gfp* (A) and controls (B) at 4 weeks after subretinal injection.

**Figure supplement 4**. IHC staining of VEGF indicates that the VEGF decreased in the area of injection (arrows) but not in the non-injected area (arrow heads).

**Figure supplement 5**. Western blot confirms Cre expression in *Vegfa$^{lox/lox}$* mice 10 days after pCre subretinal delivery (lane 1: pCre, lane 2: pNull).

**Figure supplement 6**. IL-12 levels increased in both AAV.shRNA.*sFlt-1.Gfp* and AAV.shRNA.*Gfp* treated mice (4 weeks) compared to AAV.Gfp and PBS control in the C57BL/6J mice.

To avoid the effects of mechanical disruption of the retina by subretinal injection (as shown in PBS or sham controls above), we performed conditional ablation of *Flt-1* in the RPE or photoreceptors to determine whether CNV or RAP could be induced without injury. We interbred transgenic *Vmd$_2$-cre$^+$* mice (Cre recombinase expressed specifically in RPE) (*Le et al., 2008*) or *iCre-75$^+$* mice (Cre recombinase expressed specifically in photoreceptors) (*Le et al., 2008*) with floxed *Flt-1* mice. Cre expression in the two Cre lines was identified by interbreeding with mT/mG mice, which express tomato/EGFP in all tissues (*Figure 4B*). The results showed that Cre expression, as indicated by tomato deletion resulting in exclusive *Egfp* fluorescence, was restricted to the RPE in floxed mT/mG mice bred with *Vmd$_2$-cre$^+$* and to the photoreceptors in floxed mT/mG mice crossed with the *iCre-75$^+$* line (*Figure 4B*).

At 1–3 months of age, nearly all homozygous RPE-specific *Flt-1* knockout mice (*Vmd$_2$-cre$^+$flt-1$^{lox/lox}$*) (18/22 eyes, 82%; p<0.001), and about half of the heterozygous conditional *Flt-1* knockout mice (*Vmd$_2$-cre$^+$flt-1$^{lox/+}$*) (17/42 eyes, 40%; p=0.009) developed CNV, which progressed over time, compared with 18% (2/22 eyes, 9%) of littermate controls (*Vmd$_2$-cre$^+$flt-1$^{+/+}$*). Similarly, at 1–3 months of age, both homozygous (*iCre-75$^+$flt-1$^{lox/lox}$*, 8/14 eyes, 57%; p<0.05) and heterozygous (13/24 eyes, 54%; p<0.05) photoreceptor-specific *Flt-1* knockout mice (*iCre-75$^+$flt-1$^{lox/+}$*) developed RAP, compared with 10% (1/10 eyes) of littermate controls (*iCre-75$^+$flt-1$^{+/+}$*). The number of lesions with leakages on FA ranged from one to four per eye in the floxed animals, whereas in littermate controls, there was no more than one neovascular lesion per eye. RAP lesions started from the inner retina (arising from retinal vessels) at an early stage and invaded the photoreceptor layer, eventually anastomosing with the choroidal neovasculature (*Figure 4C–E*, *Figure 4—figure supplements 1–5* and *Tables 2 and 3*).

We observed rare CNV occurrences in the littermate controls. These results may be accounted for by Cre 'leakiness' or toxicity, which has been reported in several transgenic models using the Cre-lox system (*Editorial, 2007*; *Schmidt-Supprian and Rajewsky, 2007*).

We further confirmed that sFLT-1 was down-regulated by in situ hybridization and IHC, while VEGF-A was up-regulated by western blotting in the conditional *Flt-1* knockout mice, compared with littermate controls (*Figure 4D,E*, *Figure 4—figure supplement 6*), suggesting CNV/RAP occurrence

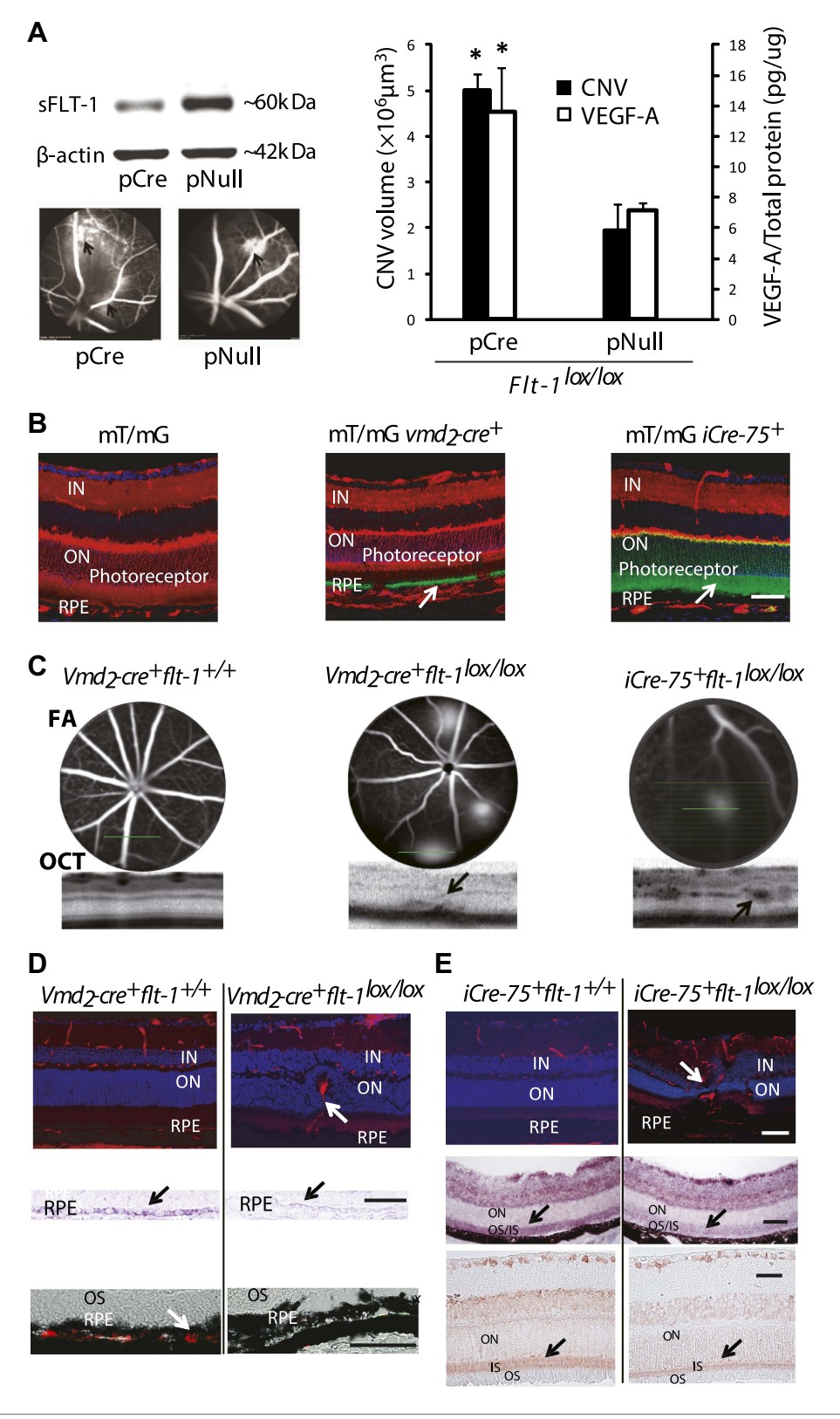

**Figure 4**. Suppression of soluble *Flt-1* by conditional *Cre/lox*-mediated *Flt-1* gene ablation induced CNV. (**A**) Western blot from the retinal pigment epithelium (RPE)/choroid tissue confirmed sFLT-1 knockdown by pCre. *Figure 4. Continued on next page*

*Figure 4. Continued*

Subretinal injection of pCre induced CNV and increased VEGF-A levels in *Flt-1$^{lox/lox}$* mice (2 weeks; arrows point to CNV). Representative FA images showed CNV lesions in both the pCre and pNull groups (left bottom). (**B**) Deletion of tomato fluorescence showed that Cre expression (mosaic pattern) was restricted to RPE in transgenic mT/mg *vmd$_2$-cre$^+$* mice (at 4 weeks of age) and was restricted to photoreceptors in transgenic mT/mG *iCre-75$^+$* mice (at 4 weeks of age), respectively (arrows point to the location of Cre expression). (**C**) Representative FA and OCT (bottom) images showed in vivo that a homozygous RPE-specific *Flt-1* knockout (*Vmd$_2$-cre$^+$flt-1$^{lox/lox}$*) mouse (48 days old) developed CNV spontaneously without surgical injury, and a homozygous photoreceptor-specific *Flt-1* knockout mouse (*iCre-75$^+$flt-1$^{lox/lox}$*) (30 days old) developed retinal angiomatous proliferation (RAP) spontaneously without surgical injury. The littermate controls without loxp were normal. (**D**) Representative IHC images (top panels) showed the retinal neovessels in the above *Vmd$_2$-cre$^+$flt-1$^{lox/lox}$* (CNV) mouse at 3 months of age. The littermate controls without loxp were normal (red stained vessels). In situ hybridization (middle panels, blue stained *sFlt-1*) and IHC images (bottom panels, red stained sFLT-1) showed decreased expression of sFLT-1 RNA and protein in the RPE (*Vmd$_2$-cre$^+$flt-1$^{lox/lox}$*) (arrows). (**E**) Representative IHC images (top panels) showed the retinal neovessels in the above *iCre-75$^+$flt-1$^{lox/lox}$* mouse (RAP) at 3 months of age. The littermate controls without loxp were normal (red stained vessels). In situ hybridization (middle panels, blue stained *sFlt-1*) and IHC images (bottom panels, red stained sFLT-1) showed decreased expression of sFLT-1 RNA and protein in the photoreceptors (*iCre-75$^+$flt-1$^{lox/lox}$*) (arrows). Scale bar: 100 μm. CNV: choroidal neovascularization; FA: fluorescein angiography; IHC: immunohistochemistry; IN: inner nuclear layer; OCT: optical coherence tomography; ON: outer nuclear layer; OS: outer segment layer.

The following figure supplements are available for figure 4:

**Figure supplement 1**. Genotyping of RPE-specific *Flt-1* knockout mice (obtained by cross-breeding with *Vmd$_2$-cre$^+$* mice and *Flt-1$^{lox/lox}$* mice) with PCR.

**Figure supplement 2**. Representative RT-PCR shows Cre expression in a *Vmd$_2$-cre$^+$ flt-1$^{+/+}$* mouse (lane 2).

**Figure supplement 3**. CNV progression in homozygous RPE-specific *Flt-1* knockout (*Vmd$_2$-cre$^+$flt-1$^{lox/lox}$*) mice.

**Figure supplement 4**. Representative toluidine blue staining images show CNV in a homozygous conditional *Flt-1* knockout mouse (*Vmd$_2$-cre$^+$flt-1$^{lox/lox}$*, 3 months old) compared to normal retinal morphology in a littermate control.

**Figure supplement 5**. Representative TEM images show CNV in a homozygous conditional *Flt-1* knockout mouse (*Vmd$_2$-cre$^+$flt-1$^{lox/lox}$*) compared to normal retinal architecture in a littermate control.

**Figure supplement 6**. Representative western blot shows increased VEGF-A levels in the RPE/choroid of a *Vmd$_2$-cre$^+$flt-1$^{lox/+}$* mouse with CNV (lane 2) compared to a littermate control (*Vmd$_2$-cre$^+$flt-1$^{+/+}$*) without CNV (lane 1).

**Figure supplement 7**. Early vascular changes with normal retinal structures occur in homozygous conditional *Flt-1* knockout mice (*Vmd$_2$-cre$^+$flt-1$^{lox/lox}$*).

**Figure supplement 8**. FA images show that no CNV has developed in homozygous conditional *Flt-1* knockout mice (*Vmd$_2$-cre$^+$flt-1$^{lox/lox}$*) at p21.

**Figure supplement 9**. Toluidine blue staining images show the morphology of the retina at p28.

is due to sFLT-1 knockdown in the RPE or photoreceptors. We observed no changes in sFLT-1 expression in the inner retinae of the two conditional *Flt-1* knockouts. However, sFLT-1 expression was somewhat decreased in the inner retina following subretinal injection with AAV.shRNA.*sFlt-1*. This may be due to diffusion and leakage of the injection fluid along the injection track.

Although there are several inherited retinal degeneration models with changes in RPE/photoreceptors where CNV does not develop, it is theoretically possible that pathological disturbances in RPE or photoreceptors due to *Flt-1* knockout may themselves contribute to the development of CNV. Furthermore, CNV by its nature is anatomically disruptive to the RPE which in turn leads to photoreceptor disruption. To determine whether the CNV/RAP lesions we observed were due to preceding retinal changes, we performed IHC staining on sections of VMD-*CreIt$^{loxp/loxp}$* mice at p7, p14, and

**Table 2.** Transgenic mice with Cre/lox-mediated conditional gene ablation of *Flt-1* in the retinal pigment epithelium (RPE) developed choroidal neovascularization (CNV) (<3 months old)

| | homozygous (*Vmd₂-cre⁺ flt-1^{lox/lox}*) | heterozygous (*Vmd₂-cre⁺ flt-1^{lox/+}*) | littermate control (*Vmd₂-cre⁺ flt-1^{+/+}*) |
|---|---|---|---|
| Litter 1 | 1/1 mice, 2/2 eyes | NA | NA |
| Litter 2 | 2/2 mice, 3/4 eyes | 5/7 mice, 9/14 eyes | 0/1 mouse 0/2 eyes |
| Litter 3 | 1/1 mouse, 2/2 eyes | 1/2 mice, 1/4 eyes | NA |
| Litter 4 | 1/1 mouse, 2/2 eyes | 2/3 mice, 2/6 eyes | NA |
| Litter 5 | 2/2 mice, 4/4 eyes | 1/2 mice, 1/4 eyes | 1/2 mice, 1/4 eyes |
| Litter 6 | 3/3 mice, 4/6 eyes | 1/2 mice, 1/4 eyes | 1/5 mice, 1/10 eyes |
| Litter 7 | 1/1 mouse, 1/2 eyes | 2/5 mice, 3/10 eyes | 0/3 mice, 1/6 eyes |
| Total | 11/11 mice (100%) | 12/21 mice (57%) | 2/11 mice (18%) |
| | 18/22 eyes (82%, p= 1.3E-6) | 17/42 (40%, p=0.009) | 2/22 eyes (9%) |

p21. No CNV lesions were seen on retinal sections at these timepoints. However, we found evidence of nascent choroidal endothelial changes abutting Bruch's membrane at p21 (*Figure 4—figure supplement 8*). At p21, the earliest feasible timepoint for angiography, we observed no CNV on FA (*Figure 4—figure supplement 9*). The earliest timepoint that we detected CNV angiographically was at p28 (*Figure 4—figure supplement 9*). Taken together, these data indicate early choroidal vascular changes without changes in the overlying neural retina.

It is important to note that we cannot exclude sFLT-1 production originating from proteolytic cleavage. To the best of our knowledge, ADAM10 is the only described protease that cleaves membrane-bound FLT-1 to the soluble isoform (*Zhao et al., 2010*). We found that ADAM10 is expressed in the inner retinal layers (*Figure 5*), but not in the RPE or photoreceptors where we found that knockdown of sFLT-1 breached photoreceptor avascular privilege. Hence, a proteolytic cleavage-mediated origin for outer retinal sFLT-1 is unlikely.

## Discussion

The neovascular processes we observed, either of CNV or RAP phenotypes in RPE/photoreceptor-specific *Flt-1* knockout mice, recapitulate many of the characteristics of human CNV or RAP (different subtypes of AMD) (*Donati et al., 2006*). In contrast to the transgenic mice which develop CNV in senility (*Ambati et al., 2003a*; *Imamura et al., 2006*; *Zhao et al., 2011*) or a model of RAP in the systemic VLDLR⁻/⁻ mouse (*Heckenlively et al., 2003*), the mouse models presented herein exhibit CNV or RAP in heterozygous transgenic mice as well and within a rapid timeframe, indicating the vital role of sFLT-1 in maintaining the integrity of photoreceptor avascular privilege. Moreover, previous

**Table 3.** Transgenic mice with Cre/lox-mediated conditional gene ablation of *Flt-1* in the photoreceptors developed retinal angiomatous proliferation (RAP) (<3 months old)

| | homozygous (*iCre-75⁺ flt-1^{lox/lox}*) | heterozygous (*iCre-75⁺ flt-1^{lox/+}*) | littermate control (*iCre-75⁺ flt-1^{+/+}*) |
|---|---|---|---|
| Litter 1 | 2/4 mice, 3/8 eyes | NA | 0/1 mouse 0/2 eyes |
| Litter 2 | 1/1 mice, 1/2 eyes | 3/4 mice, 6/8 eyes | NA |
| Litter 3 | 1/1 mouse, 2/2 eyes | 2/4 mice, 4/8 eyes | 0/1 mouse, 0/2 eyes |
| Litter 4 | 1/1 mice, 2/2 eyes | 1/2 mice, 1/4 eyes | 1/2 mice, 1/4 eyes |
| Litter 5 | NA | 1/1 mouse, 2/2 eyes | NA |
| Litter 6 | NA | 0/1 mouse, 0/2 eyes | 0/1 mouse, 0/2 eyes |
| Total | 6/7 mice (86%) | 7/12 mice (58%) | 1/5 mice (20%) |
| | 8/14 eyes (57%, p<0.05) | 13/24 (54%, p<0.05) | 1/10 eyes (10%) |

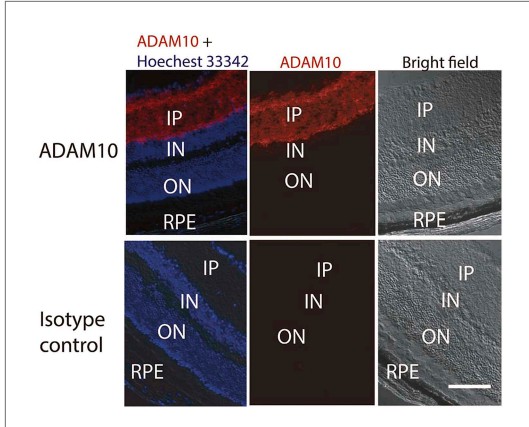

**Figure 5**. Representative IHC staining images show the ADAM10 expression in the inner layer of the retina. IHC: immunohistochemistry; IN: inner nuclear layer; IP: inner plexiform layer; ON: outer nuclear layer; RPE: retinal pigment epithelium.

studies found that transgenic mice which overexpress VEGF-A in either RPE or photoreceptors developed intrachoroidal neovascularization or retinal neovascularization but not CNV or RAP (*Okamoto et al., 1997*; *Schwesinger et al., 2001*). These phenomena suggest that overexpression of VEGF-A in the retina or RPE is not sufficient to overcome the subretinal vascular barrier to induce CNV. On the other hand, in our study, mice with reduced sFLT-1 expression in the RPE developed CNV, indicating that sFLT-1 down-regulation is sufficient to cause CNV. Hence, the ratio of soluble FLT-1 to VEGF is likely critical to the maintenance of the subretinal vascular barrier. We believe physiologic sFLT-1 is not disruptive to normal neural function due to both a critical balance and the polarity of sFLT-1 expression by RPE. Further, the models presented herein more closely mimic human AMD, rendering these transgenic models quite useful for further studies and translational applications.

While diffusion of sFLT-1 between the RPE and photoreceptors is theoretically possible, the intervening interphotoreceptor matrix could trap sFLT-1, preventing free diffusion. In addition, sFLT-1 secretion by the RPE is polarized basally towards the choroid; therefore, VEGF secretion from the RPE can elicit choroidal neovascularization.

We observed less CNV in the injection site after AAV.shRNA.*sFlt-1* delivery than after antibody or genomic knockdown; this may be due to persistence of preformed sFLT-1, as the turnover of this protein in the RPE is, to our knowledge, unknown. Another interesting observation is that lower levels of both VEGF-A and CNV were observed in *Vegfa*[lox/lox] mice compared with wild type mice in AAV-mediated sFlt-1 knockdown studies. It is possible that the insertion of lox sites flanking the *VEGF-A* gene can affect gene expression in conditions of injury. This could be due to several mechanisms including disruption of cis-regulatory elements, interruption of unknown miRNA binding sites, and effects on mRNA stability by the palindromic nature of loxp sites. Loxp insertion has previously been reported to affect the transcription of genes in cis in selected circumstances (*Sun and Storb, 2001*; *Meier et al., 2010*).

Nonetheless, taken together, our findings demonstrate that protein blockade, transcriptional silencing, or genomic ablation of sFLT-1 can induce CNV/RAP, showing that sFLT-1, through sequestration of VEGF-A, is a key molecular defender of vision-critical photoreceptor avascular privilege. Our transgenic models would not be expected to directly affect *Flt-1* expression in retinal or choroidal vessels. However, the knockdown of *Flt-1* in photoreceptors in our *iCre-75*[+]*flt-1*[lox/lox] mice may contribute to focal areas of photoreceptor degeneration due to the depletion of *Flt-1* compromising Wnt signaling.

Reduced VEGF-A expression has been observed in CNV specimens surgically obtained from patients with persistent CNV not responsive to anti-VEGF-A therapy. This reduction may explain why some patients stop responding to anti-VEGF-A therapy after a period of time (*Tatar et al., 2009*). Endothelial sFLT-1 expression is regulated by VEGF, acts as an autocrine regulator of endothelial cell function, and controls angiogenesis (*Ahmad et al., 2011*). An important area of future investigation will be to determine, based on stage of CNV development, how anti-VEGF-A therapy can be complemented both with approaches to suppress underlying hypoxia and inflammation and with strategies to maintain or increase sFLT-1 expression, perhaps revitalizing normal vascular barriers.

The milieu in the photoreceptor/RPE/Bruch's membrane/choriocapillaris is complicated. Some endogenous angiogenesis inhibitors (e.g., pigment epithelium-derived factor) in the RPE/Bruch's membrane/choriocapillaris complex have been reported to be reduced in AMD and may contribute to the barrier against choroidal neovascular invasion (*Bhutto et al., 2008*). Nevertheless, the findings in this study offer greater insight into the molecular mechanisms responsible for retinal vascular demarcations and CNV/RAP, and the described molecular advances have broader ramifications. They impact testing of anti-angiogenic pharmaceuticals, may illuminate the mechanistic basis of angiogenesis in other

diseases affecting over a billion people (*Potente et al., 2011*), hold translational import for a key cause of blindness, and support the ongoing clinical trial of AAV.*sFlt-1* for the treatment of neovascular AMD (clinical trial NCT01024998 at www.clinicaltrials.gov). Further, they highlight a potential use of sFLT-1 as a therapeutic target in situations where angiogenesis is desirable, as well as a new role for sFLT-1, which has recently been found to protect renal podocytes (*Jin et al., 2012*), and the genetic models of AMD we describe are the first which exhibit CNV or RAP spontaneously at an early age, rendering them original non-invasive platform technologies with a variety of applications.

## Materials and methods

### Mice

C57BL/6J mice (6–8 weeks; The Jackson Laboratory, Bar Harbor, ME), Balb/cJ mice (6 weeks; The Jackson Laboratory), *Vegfa$^{lox/lox}$* mice (*Gerber et al., 1999*) (6–8 weeks, C57BL/6 background; gift of Genentech, San Francisco, CA), *Tlr3$^{−/−}$* mice (6–8 weeks, B6;129S1-*Tlr3$^{tm1Flv}$*/J; The Jackson Laboratory), *Flt-1$^{lox/lox}$* mice (*Baldwin et al., 2004*) (6–8 weeks, gift of Genentech), *Vmd$_2$-cre* mice (*Le et al., 2008*) (8–10 weeks, FVB/N mice background, heterozygotes do not have any RD1 mutation; Cre expression in RPE continues from embryonic day 9 to postnatal day 60; gift of Dr Yun Zheng Le, University of Oklahoma Health Sciences Center), *iCre-75$^+$* mice (*Li et al., 2005*) (14–20 weeks, C57BL/6 × SJL background; Cre expression in photoreceptors begins earlier than postnatal day 11; gift of Dr Ching-Kang Chen, University of Utah), mT/mG mice (14–20 weeks; STOCK *Gt(ROSA)26Sor$^{tm4(ACTB-tdTomato,-EGFP)Luo}$*/J; The Jackson Laboratory) and *Flt-1Tk$^{−/−}$* mice (14–20 weeks) (*Hiratsuka et al., 1998*) were used. Experimental groups were age and sex matched. To identify Cre expression in the RPE layer of *Vmd$_2$-cre$^+$* (*Le et al., 2008*) or photoreceptor layer of *iCre-75$^+$* mice (*Li et al., 2005*), the mT/mG mice which express tomato/EGFP in all tissues were interbred with the above lines, respectively and the deletion of tomato fluorescence was checked at the age of 4 weeks. To generate RPE (or photoreceptor) specific *Flt-1* knockout mice, *Vmd$_2$-cre$^+$* (or *iCre-75$^+$*) mice and *Flt-1$^{lox/lox}$* mice were cross-bred to obtain the homozygous (*Vmd$_2$-creflt-1$^{lox/lox}$* or *iCre-75$^+$ flt-1$^{lox/lox}$*), heterozygous (*Vmd$_2$-cre$^+$ flt-1$^{lox/+}$* or *iCre-75$^+$ flt-1$^{lox/+}$*), and littermate controls (*Vmd$_2$-cre$^+$flt-1$^{+/+}$* or *iCre-75$^+$flt-1$^{+/+}$*). These mice were identified with genotyping as described (*Li et al., 2005*; *Le et al., 2008*).

All mice were age and sex matched (mice used for subretinal injection were 6–8 weeks old and male) and fed with autoclaved 8656 Teklad Sterilizable Rodent diet provided by Harlan Laboratories (Indianapolis, IN). Cages were routinely changed once every 2 weeks throughout the course of this study. Experiments were approved by the University of Utah Institutional Animal Care and Use Committee and conformed to the Association for Research in Vision and Ophthalmology Statement on Animal Research.

### Human samples

Normal human fresh globes were obtained within 4 hr of death from the Utah Lions Eye Bank, and sections were prepared as described below. Cryosection slides (12 µm or 5 µm thickness) of human ocular AMD and normal control tissue were obtained from the Ambati Laboratory at the University of Kentucky. AMD tissue and normal human tissue was matched for gender and age in each case. Specimens were taken from the eyes of AMD patients who had never received anti-VEGF antibody (Lucentis or Avastin) therapy.

### Vectors

AAV.shRNA.*sFlt-1* was developed as previously described (*Ambati et al., 2006*). In brief, short hairpin (sh)RNA expression cassettes (SECs) were developed by in vitro PCR amplification. SECs are PCR products that consist of promoter and terminator sequences flanking a hairpin siRNA template. Multiple target sequences along with different combinations of promoters were screened to identify the most effective siRNA capable of gene knockdown in vitro and in vivo. We used pAAV-MCS plasmids (AAV Helper-Free System; Agilent Technologies, Santa Clara, CA) containing inverted terminal repeats (*Line et al., 2005*). We obtained Ac*Gfp* cDNA from pIRES2-Ac*Gfp1* (Clontech Laboratories, Mountain View, CA), and generated pAAV-Ac*Gfp* to insert Ac*Gfp* cDNA into the multi-cloning site of pAAV-MCS. sh*Flt* and shNegative cDNA with an H1 promoter were inserted into pAAV-Ac*Gfp* using a PstI site to obtain pAAV-Ac*Gfp*-sh*Flt* and pAAV-Ac*Gfp*-shNegative. Serotype 2 AAV vector containing plasmids were produced by the Viral Vector Core at the University of Massachusetts Medical School. The target (sFLT-1) sequence is aatgattgtaccacacaaagt. A target for shRNA in identical regions of human and

mouse soluble FLT-1 unique intron 13 derived tail was selected, since the intron 13 derived tail of 31 amino acids is only present in soluble FLT-1 and not in membrane FLT-1, so soluble FLT-1 was specifically abrogated.

## Locked nucleic acid probes

A 21 base DNA oligonucleotide with 30% locked nucleic acid (LNA; Exiqon, Woburn, MA) substitutions was used as an anti-sense probe against sFLT-1 mRNA. The sequence of the LNA probe was actttgtgtggtacaatcatt. The calculated RNA Tm value is 46°C. A 21-mer sFLT-1-sense-LNA probe with a sequence aatgattgtaccacacaaagtc (predicted RNA Tm value of 46°C) was used as control. Both LNA probes were digoxigenin (*Kaneko et al., 2011*) labeled at the 5′ and 3′ ends.

## Subretinal injection

Mice were placed under general anesthesia with an intraperitoneal injection of tribromoethanol (289 mg/kg). Topical 0.5% proparacaine solution and 1% tropicamide ophthalmic solution obtained from the Moran Eye Center pharmacy were applied to the cornea as an anesthetic and dilator, respectively. Using an Olympus stereomicroscope for magnification, a small incision was made behind the limbus with a 30.5 gauge needle. To deliver 1 μl of each treatment, a custom 33 gauge, blunt tipped microsyringe (Hamilton, Reno, NV) was inserted through the incision, vitreous, and retina into the subretinal space in the posterior pole of the retina, taking care to avoid the lens. Fundus examinations during the injection procedure showed partial retinal detachment, confirming successful subretinal delivery.

To inhibit sFLT-1 by shRNA in C57BL/6J mice, 1 μl of AAV.shRNA.s*Flt-1.Gfp* ($1.4 \times 10^{11}$ genomic copies/ml) was injected subretinally, with AAV.shRNA.*Gfp* ($1.4 \times 10^{11}$ genomic copies/ml), AAV.*Gfp* ($1.4 \times 10^{11}$ genomic copies/ml) or PBS injected as controls.

For genomic deletion in *Flt-1^{lox/lox}* mice, a plasmid encoding Cre recombinase (pCre; 1 μg, gift of RK Nordeen, University of Colorado) with 10% NeuroPORTER Transfection Reagent (Genlantis, San Diego, CA) as a plasmid transfection enhancer (*Kachi et al., 2005*) was injected into the subretinal space. Analogous empty plasmid (pNull, 1 μg) with 10% NeuroPORTER was injected as a control.

For genomic deletion in *Vegfa^{lox/lox}* mice, 1 μl of 50% pCre, 10% NeuroPORTER, and 40% AAV.shRNA.s*Flt-1.Gfp* was injected into the subretinal space. As controls, 1 μl of the following vectors were injected into the subretinal space of *Vegfa^{lox/lox}* mice: (1) 50% pNull, 10% NeuroPORTER, and 40% AAV.shRNA.s*Flt-1.Gfp*; (2) 50% pCre, 10% NeuroPORTER, and 40% AAV.shRNA.*Gfp*; (3) 50% pCre, 10% NeuroPORTER, and 40% AAV.*Gfp*; or (4) 50% pCre, 10% NeuroPORTER, and 40% 1× PBS.

A 1 μl (0.2 μg) sample of goat FLT-1 antibody (mouse myeloma cell line NSO derived recombinant mouse VEGFR1, Ser27Glu759; R&D Systems, Minneapolis, MN) or equivalent goat isotype IgG (Jackson ImmunoResearch, West Grove, PA) as control were subretinally injected into C57BL/6J, *Flt-1tk^{−/−}*, or *Tlr3^{−/−}* mice, respectively.

## CNV observation in vivo

Mice were anesthetized with tribromoethanol (289 mg/kg) and the pupils were dilated with topical 1% tropicamide ophthalmic solution (Bausch & Lomb, Rochester, NY). All in vivo imaging described was performed using a Spectralis confocal ophthalmoscope (Heidelberg Engineering, Vista, CA). HRA-OCT imaging of mouse fundi was performed 2, 4, 6, and 10 weeks after subretinal injections with AAV.shRNA.s*Flt-1*, or at 21 days, 1, 2, and 3 months of age in the transgenic mice. Green fluorescent protein (GFP) expression was observed with a red free or autofluorescence filter. Mice were intraperitoneally injected with 0.1 ml of 10% sodium fluorescein (Akorn, Lake Forest, IL) or 0.2 ml of 8 mg/ml indocyanine green (ICG, Akorn). CNV was evaluated by experienced retinal specialists using a combination of images obtained by FA, optical coherence tomography (OCT), ICG, or a MICRON II small animal retinal imaging microscope (Phoenix Research Laboratories, San Ramon, CA).

## Computer-assisted quantitative analysis of OCT images

Quantification of lesions in vivo was performed by using OCT images according to established procedures (*Schmucker and Schaeffel, 2004*; *Joeres et al., 2007*; *Furino et al., 2009*; *Ruggeri et al., 2009*; *Giani et al., 2011*; *Luo et al., 2013*). This method has been widely used for estimating CNV volumes in vivo, especially in AMD patients, and has proved to be a highly reproducible quantitative measurement (*Joeres et al., 2007*; *Giani et al., 2011*; *Luo et al., 2013*). B-scans were exported from Heidelberg Eye Explorer (HEE), Heidelberg's image analysis software, in .jpeg format and imported as complete stacks

into Seg3D software. Seg3D is a segmentation processing and analyzing tool developed by the University of Utah Scientific Computing and Imaging Institute and the NIH/NCRR Center for Integrative Biomedical Computing (Volumetric Image Segmentation and Visualization, Scientific Computing and Imaging Institute [*Furino et al., 2009*], available at http://www.seg3d.org). Boundaries of the CNV lesion and deformations of the RPE were manually outlined using a polyline tool by a masked and experienced technician (*Luo et al., 2013*; *Yang et al., 2013*). Seg3D masked the region enclosed within the polyline for each consecutive b-scan in pixels squared. An automated feature of Seg3D calculated two-dimensional lesion area, summing the masked b-scans for each stack. Scaling factors obtained from HEE were used to convert $pixel^2$ to $\mu m^2$. After adjusting for murine corneal curvature to 1.4 mm (*Ahlers et al., 2008*), the scaling factors for both x- and z-axes were 1.45 ± 2 and 3.87 μm/pixel, respectively. To calculate lesion volume ($\mu m^3$), the lesional area on each scan line was multiplied by the distance between scans (52 ± 2 μm; provided by HEE software), and then summated. Finally, estimated volumes were converted to $mm^3$ and reported.

CNV was quantified by FA (at 3–5 min after fluorescein injection) and OCT 10 days after antibody injection and 4 weeks after injection of AAV.shRNA.*sFlt-1*.

## Cryosection

The eyes were enucleated, fixed in 4% paraformaldehyde (PFA) (Electron Microscopy Sciences, Hatfield, PA) for 2 hr at 4°C, cryoprotected in 30% sucrose overnight, and embedded in Tissue-Tek O.C.T. Compound (Sakura Finetek USA, Torrance, CA). Sections (12 μm) were cut on a cryostat.

## Immunohistochemistry

Rabbit sFLT-1 primary antibody (*Orecchia et al., 2003*) (0.03 μg/μl, rabbit anti-C-terminal of sFLT-1) was provided by Drs Lacal and Orecchia, Instituto Dermopatico dell'Immacolata, Instituto di Ricovero e Cura a Carattere Scientifico (IDI-IRCCS), Rome, Italy. The same concentrations of rabbit IgG (Invitrogen, Carlsbad, CA) were used as isotype controls. The Vector M.O.M. (Mouse on Mouse) immunodetection kit (Vector Laboratories, Burlingame, CA) was used to detect mouse antibody on mouse tissue according to the manufacturer's instructions. To quench autofluorescence, 0.3% Sudan Black (wt/vol) (Sigma-Aldrich, St. Louis, MO) in 70% EtOH (vol/vol) was applied to slides; Dylight 405 (1:100; Jackson ImmunoResearch) was used as a secondary antibody to avoid RPE autofluorescence without Sudan Black application.

To determine sFLT-1 and/or VEGF expression in mice, sections were blocked in 10% goat serum (Abcam, Cambridge, MA) diluted in IHC buffer (1% BSA, 1% FBS, and 0.3% Triton X-100 in PBS pH 7.4) for 1 hr followed by overnight incubation with rabbit sFLT-1 primary antibody and/or mouse anti-VEGF monoclonal antibody (1:100; Abcam) diluted in blocking solution. The Vector M.O.M. immunodetection kit was used to detect mouse antibody on mouse tissue according to the manufacturer's instructions. After primary antibody incubation, slides were washed and incubated in secondary antibody Alexa Fluor 488 goat anti-rabbit (1:400; Invitrogen) and/or Alexa Fluor 546 goat anti-mouse (1:400; Invitrogen) for 1 hr at room temperature (RT). After washing three times with PBS, VECTASHIELD Mounting Medium with DAPI (Vector Laboratories) was applied and slides were coverslipped. The same concentrations of rabbit IgG were used as isotype controls.

To determine sFLT-1 and/or VEGF expression in human tissue, the same primary secondary antibodies were used as above. After secondary antibody application, 0.3% Sudan Black (wt/vol) was applied to slides for 10 min to quench autofluorescence. Slides were rinsed quickly with PBS eight times and mounted. Dylight 405 was used instead of the above secondary antibody to avoid the RPE autofluorescence without Sudan Black application.

To demonstrate vessels in the human sections, rat anti-heparan sulfate proteoglycan (perlecan) monoclonal antibody (1:200, clone A7L6; Millipore, Billerica, MA) was used as primary antibody, and anti-rat IgG-FITC (1:1000, F6258; Sigma-Aldrich) was used as secondary antibody, followed by Sudan Black staining as describe above. VECTASHIELD Mounting Medium with DAPI was used and the slides were coverslipped.

## Histology

Hematoxylin and eosin (H&E) staining, toluidine blue staining, and transmission electron microscopy followed the protocols (http://prometheus.med.utah.edu/~marclab/marclab_09_tools-protocols.html) (*Luo et al., 2013*). Samples were sectioned at 1 μm for H&E staining or toluidine blue staining. The following materials were used: 10% neutral buffered formalin (EMD Chemicals USA, Gibbstown, NJ),

hematoxylin solution (Sigma-Aldrich), acid alcohol (Thermo Scientific, Waltham, MA), Bluing Reagent (0.25 NH$_4$ solution; Richard-Allan Scientific, Kalamazoo, MI), Eosin solution (Sigma-Aldrich), and Cytoseal XYL (Fisher Scientific, Pittsburgh, PA).

## Toluidine blue staining

Sections were cut at 1 µm, transferred with a drop of distilled water onto a glass slide, and then dried on a slide warmer. After the sections were completely dried, they were covered with a few drop of toluidine blue solution (0.05% toluidine blue and 1% sodium borate in distilled water, mixed at a ratio of 1:2 with the heat source still on) for 1–2 min, then excess solution was soaked up and the slide was air-dried, followed by coverslipping with the above mounting medium.

## ADAM10 staining

Cryosections (12 µm) from day 3 of laser induced CNV murine eyes were incubated at 37°C for 40 min. After washing with PBS twice, the sections were incubated in 0.25% Triton X-100/PBS for 15 min at RT. After washing with PBS twice, the sections were blocked in 5% donkey serum/0.02% Triton X-100/PBS for 30 min at RT. Primary antibody (Abcam ab1997 for ADAM10; Invitrogen 02-6102 for isotype) was applied to the section at 5 µg/ml in blocking buffer, and incubated for 1 hr at RT. After washing with PBS five times, 1:1000 Alexa546 conjugated donkey anti-rabbit IgG Fab fragment (Invitrogen A11071) was applied to the section, and incubated for 30 min at RT. After washing with PBS three times, the nucleous was stained with Hoechest33342 (Invitrogen) and the sections were then mounted. Images were taken with an inverted fluorescence microscope (*Klein et al., 2005*).

## Transmission electron microscopy

Fixed tissues were osmicated for 45–60 min in 0.5–1% OsO$_4$ in 0.1 M cacodylate buffer, processed in maleate buffer for staining with uranyl acetate, and resin-embedded. Ultrathin sections were cut at 60 nm with a Leica ultramicrotome and imaged at 80 KeV on a JEOL JEM 1400 electron microscope. Images were captured with a 16-megapixel GATAN Ultrascan 4000 camera. All fluorescent sections and H&E stained sections were imaged with an Axiovert 200 microscope (Carl Zeiss MicroImaging, Thornwood, NY) equipped with confocal epifluorescence illumination or an AxionCamMR digital camera (Carl Zeiss Microscopy, Thornwood, NY) and processed using Adobe Photoshop software (Adobe Systems, San Jose, CA). Unstained tissue sections were mounted and coverslipped and then directly examined under the above microscopes.

## In situ hybridization

Cryosections (12 µm) of BALB/cJ mouse eye tissue were prepared as described above. Sections were fixed in 4% PFA for 10 min followed by 5 µg/ml proteinase-K treatment in DEPC treated water at RT for 10 min. Pre-hybridization was performed in hybridization buffer (50% formamide, 5× SSC, 5× Denhardt's, 250 µg/ml yeast RNA, 500 µg/ml SSD, 2% Roche blocking reagents, and 0.2% Tween 20) at RT for 4 hr, followed by hybridization with 20 nM anti-sense sFLT-1 LNA probe. Hybridization was performed at 55°C. After hybridization, stringent washes were performed at 60°C with 5× SSC for 30 min, 2× SSC for 30 min, 1× SSC for 30 min, and 0.2× SSC for 1 hr. Procedures to avoid RNase contamination were used during the following steps. Slides were blocked in Tris buffer (0.1 M Tris pH 7.5/0.15 M NaCl) containing 10% sheep serum for 1 hr and then incubated in alkaline phosphatase-conjugated anti-digoxigenin antibody (1:2000; Roche Applied Science, Indianapolis, IN) overnight at 4°C. Enzymatic development was performed using 4-nitroblue tetrazolium (NBT) and 5-brom-4-chloro-30-indolylphosphate (BCIP) substrate (Roche Applied Science), forming a dark blue NBT-formazan precipitate within 2–3 hr. Images were obtained with an Axiovert 200 inverted microscope (Carl Zeiss Microscopy). In order to avoid the interruption from dark RPE, white mice lines were used.

## Protein expression

Mouse RPE and choroid were harvested, placed in 10% RIPA buffer (Sigma-Aldrich) and sonicated 3–5 times. Samples were spun at 14,000×*g* for 15 min and the supernatant, containing total protein extract, was transferred to a clean labeled tube. Tissue supernatant was used to perform all protein assays. Total protein was quantified using a Pierce BCA Protein Assay Kit (ThermoFisher Scientific, Rockford, IL) according to the manufacturer's protocol. Mouse VEGF-A (BMS619; eBioscience, San Diego, CA), mouse IL-12 (KMC0121; Invitrogen), mouse IFN-γ (KMC4021; Invitrogen), and mouse placental growth factor (E0114m; Uscn Life, Wuhan, China) ELISAs were performed according to the

manufacturer's instructions to quantify VEGF, IL-12 and IFN-γ, and PlGF, respectively. All assays were read using an ELx800 Absorbance Microplate Reader and Gene5 software (BioTek, Winooski, VT).

## SDS-PAGE and western blot

AAVs or plasmid subretinal injections are described above. Two weeks after the injection, we examined sFLT-1 protein reduction by AAV.shRNA.*sFlt-1* using western blot from the RPE/choroid. After the mice were euthanized, the RPE/choroid complex was separated from the sclera under a stereomicroscope and placed in 120 µl of RIPA buffer (Sigma-Aldrich) containing protease inhibitor cocktail (Roche Applied Science). Samples were homogenized with a sonic dismembrator (ThermoFisher Scientific), and total protein concentration was determined using 10 µl of sample and a Pierce BCA assay kit (ThermoFisher Scientific). The proteins were eluted with Laemmli buffer (Bio-Rad Laboratories, Hercules, CA) on ice for 5 min, boiled for 5 min, and centrifuged at 14,000×*g* for 5 min. Then 10 µg of protein were run through a 10% acrylamide gel under reducing conditions. After SDS-PAGE, proteins were transferred to a nitrocellulose membrane and incubated for 1 hr with blocking buffer (3% BSA and 0.05% Tween 20 in TBS for FLT-1 or 5% non-fat milk and 0.05% Tween 20 in TBS for other proteins) then incubated with anti-FLT-1 antibody targeting D1–D5 of FLT-1 (1:1000, ab9540; Abcam) which recognize sFLT-1 and mFLT-1, or Novagen anti-CRE antibody (1:5000, 69050; EMD Millipore, Billerica, MA) overnight at 4°C. The membrane was washed once in TBST (0.05% Tween 20/TBS) and twice in TBS. Finally, the membrane was incubated with the appropriate secondary HRP-linked antibody in blocking buffer for 30 min at RT. After washing once with TBST and three times with TBS, the membrane was developed using an Amersham ECL Plus Western Blotting Detection Kit (GE Healthcare Bio-Sciences, Piscataway, NJ). Chemiluminescence was detected by a FOTO/Analyst Electronic Imaging System (FOTODYNE, Hartland, WI). As an internal control, a western blot detecting β-actin was performed as follows: the membrane was incubated in Pierce Restore Plus Western Blot Stripping Buffer (ThermoFisher Scientific) followed by blocking, primary antibody incubation, washing, HRP conjugated secondary antibody incubation, washing, and development as described above. Mouse monoclonal anti-β-actin (1:5000, ab6276; Abcam) was used as the primary antibody.

## Statistical analysis

Differences in CNV volumes and mean level of protein or mRNA were compared using the Student t test or Mann–Whitney U test with Bonferroni correction for multiple comparisons. Differences in CNV occurrence in transgenic mice were compared with the $\chi^2$ test. Statistical significant was set at $p < 0.05$. Data are presented as mean ± SEM.

## Acknowledgements

We are grateful to the patients who donated their eyes to science. We are especially indebted to Bonnie Archer, who was extremely helpful and offered invaluable assistance, support and management expertise. We thank Drs Zong-Zhong Tong and Valeria Tarallo for thoughtful discussions and critical reading of the manuscript; Dr Kristen Zygmunt and Ross Whitaker for their help with Seg3D software; Ying Liu, Alex D Jones, Dr Robert Marc, Jeanne Frederick, and James Anderson for technical support; and Ching-Kang Chen for transgenic mice. We also acknowledge the Italian Ministry of Health for financial support in the production of the sFLT-1 antibody and the unrestricted awards by Research to Prevent Blindness to the Moran Eye Center and Casey Eye Institute. The content is solely the responsibility of the authors and does not necessarily represent the official views of the National Institutes of Health.

## Additional information

### Funding

| Funder | Grant reference number | Author |
|---|---|---|
| National Eye Institute | NEI 5R01EY017950 | Balamurali K Ambati |
| RPB Physician-Scientist Award | | Balamurali K Ambati |
| National Eye Institute | NEI 5R01EY20900 | Balamurali K Ambati |

| Funder | Grant reference number | Author |
|---|---|---|
| American Diabetes Association | ADA 1-10-BS-94 | Yun Zheng Le |
| VA Merit Award | | Balamurali K Ambati |
| Beckman Initiative for Macular Research | Grant 1003 | Balamurali K Ambati |
| Doris Duke Charitable Foundation | | Jayakrishna Ambati |
| Burroughs Wellcome Fund | | Jayakrishna Ambati |
| Ellison Medical Foundation | | Jayakrishna Ambati |
| National Eye Institute | NEI R01EY022238 | Jayakrishna Ambati |
| National Eye Institute | NEI R01EY020672 | Jayakrishna Ambati |
| Programme for Advanced Medical Education sponsored by Fundação Calouste Gulbenkian, Fundação Champalimaud, Ministério da Saúde, and Fundação para a Ciência e Tecnologia, Portugal | | Ana Bastos-Carvalho |
| National Natural Science Foundation of China | 81271016 | Ling Luo |

The funders had no role in study design, data collection and interpretation, or the decision to submit the work for publication.

## Author contributions

LL, KT, Conception and design, Acquisition of data, Analysis and interpretation of data, Drafting or revising the article; HU, SKD, TO, JMS, KJ, NS, TRM, WH, FA, AB-C, CM, JB, KY-h, LO, Acquisition of data, Analysis and interpretation of data, Drafting or revising the article; XZ, Acquisition of data, Analysis and interpretation of data; DH, Conception and design, Acquisition of data, Analysis and interpretation of data; YZL, Analysis and interpretation of data, Drafting or revising the article, Contributed unpublished essential data or reagents; VAC, Acquisition of data, Analysis and interpretation of data, Contributed unpublished essential data or reagents; WWH, MS, SG, Conception and design, Acquisition of data, Drafting or revising the article, Contributed unpublished essential data or reagents; PML, AO, NF, Acquisition of data, Analysis and interpretation of data, Drafting or revising the article, Contributed unpublished essential data or reagents; GG, DYL, KVC, Conception and design, Analysis and interpretation of data, Drafting or revising the article, Contributed unpublished essential data or reagents; YF, WB, Conception and design, Acquisition of data, Contributed unpublished essential data or reagents; RA, Conception and design, Acquisition of data, Drafting or revising the article; DJW, Conception and design, Analysis and interpretation of data, Contributed unpublished essential data or reagents; JA, Conception and design, Acquisition of data, Analysis and interpretation of data, Drafting or revising the article, Contributed unpublished essential data or reagents; BKA, Conception and design, Acquisition of data, Analysis and interpretation of data, Drafting or revising the article, Contributed unpublished essential data or reagents

## Ethics

Animal experimentation: This study was performed in strict accordance with the recommendations in the Guide for the Care and Use of Laboratory Animals of the National Institutes of Health. All animals were handled according to University of Utah IACUC approved protocol # 10-07013. All procedures were performed under ketamine/xylazine, and every effort was made to minimize suffering.

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
