## [Decision Letter]

Thank you for choosing to send your work entitled ‘Photoreceptor Avascular Privilege is shielded by soluble VEGF receptor-1’ for consideration at *eLife*. Your article has been evaluated by a Senior editor and 3 reviewers, one of whom is a member of our Board of Reviewing Editors.

The Reviewing Editor and the other reviewers discussed their comments before we reached this decision, and the Reviewing Editor has assembled the following comments based on the reviewers' reports.

Luo et al. present a series of experiments that suggest that soluble VEGFR1 (*sFlt-1*) is responsible for the avascularity of the photoreceptors and subretinal space. This group had previously published a study that linked *sFlt-1* expression in the cornea to the avascularity of the cornea. In order to test the normal function of *sFlt-1* the authors have blocked *sFlt-1* via three different, independent approaches: anti-sFLT-1 antibody, *sflt-1* conditional knockouts, and *sflt*-siRNA. Each one of the three approaches has its weaknesses, but the fact that each approach has the same outcome makes the overall study quite compelling. The model created by eliminating *sFlt-1* is of general interest since a simple KO of RPE *sFlt-1* yields a reproducible choroidal neovascular (CNV) response, i.e., a model for wet AMD without surgical intervention. Even more interesting is the observation that conditional KO of *sFlt-1* in RPE causes CNV whereas a conditional KO of *sFlt-1* in photoreceptors yields retinal angiomatous proliferation (RAP). The authors suggest that reduced levels of *sFlt-1* in the retina may be linked to diseases where blood vessels invade the photoreceptor layer, such as AMD.

The consensus of the reviewers is that this is an important study with significant clinical implications, but in its present form the manuscript has a number of issues, both experimentally and in terms of the level of detail in the writing. Both need to be addressed before it is suitable for publication.

1) First, we would like to make a philosophical point. The authors should ensure that their central claim rests on experiments that are objectively reproducible or falsifiable. This field has been beset by data that do not meet that bar. For example, if other scientists fail to reproduce the subretinal injection experiments in which the outcome is CNV volume (e.g., Figure 3), the authors could claim that special skill is required to do the micro-injection correctly. Therefore, the reviewers put more weight on experiments that require no special manipulation, in particular the RPE-specific and photoreceptor-specific *Flt-1* KOs. Other scientists should be able to breed these lines of mice, perform the histology, and see the neovascularization. If other scientists do not obtain the same result, then that would falsify the central claim of this paper. For this reason, the authors should present a quantification of the neovascularization events per retina for this particular set of experiments. Examples are shown in Figure 4 but one cannot tell whether this represents a common or rare occurrence in these eyes.

2) Regarding RPE vs photoreceptor *sFlt-1* loss, it is hard to believe that loss of *sFlt-1* from one cell type would not be compensated for by diffusion of sFLT-1 made by the other cell type.

3) It is not clear how exactly CNVs were quantified. We have the impression OCT was used as the main readout for CNV, which is questionable. Or was fluorescein or ICG data included in the quantification? If yes, how was this done? And how exactly was the OCT data quantified? Looking at the OCTs in Figure 1, we would struggle to quantify changes here. Most importantly, how can CNVs be distinguished in this image modality from fluid build-up due to leakage? Why have the authors not used IHC to visualize the CNVs in whole mounts? This would make the CNV quantification data much more reliable. A similar criticism applies to Figure 3. Why have the authors not used a vessel specific stain to show CNVs? It is impossible to be certain that vascular changes have occurred based on the H&E stains shown. The ‘V’ in CNV stands for ‘vascularization’, yet the authors do not show any vascular specific stains (apart from Figure 4). This is a major weakness.

4) Figure 2: the authors should more carefully characterize how *sFlt-1*, the blocking antibody, VEGF, and the anti VEGF antibodies may affect their VEGF and PlGF quantification. For instance, ‘free VEGF’ is mentioned. How can the authors be sure that their ELISA does not measure ‘non-free’ VEGF (bound to *sFlt-1*)? The authors seem to imply that the *sFlt-1* blocking antibody can displace VEGF from bound *sFlt-1* (thereby raising VEGF levels). This is a bit surprising, considering the very high affinity *sFlt-1* has for VEGF. The authors should demonstrate that this is possible for VEGF (but not for PlGF) in vitro.

5) Figure 2: how can the authors explain the fact that PlGF is not changed? Similar questions as above arise. Is the anti-*sFlt-1* blocking antibody not capable of displacing PlGF (but it can do that with VEGF)? Also, can the authors be sure that they are measuring true ‘free PlGF’ (and not PlGF bound to *sFlt-1*)?

6) Figure 3: the authors claim their shRNA approach does not affect *mFlt-1*. Why have they not shown this in the Western blot in panel A (e.g., instead of β-actin)? This would be much more convincing.

7) Figure 3: why does the *sFlt-1* IHC look so completely different in Figure 3 compared to Figure 1? In Figure 3 the RPE has similar intensity to the IS, whereas in Figure 1 the RPE is much weaker.

8) Figure 3: how far towards the inner retina is *sFlt-1* reduced? The authors show in Figure 3 only a very small part of the retina that does not include the IN, ON, and RGC layer. However, Figure 1 suggests that there is significant expression in these locations. Have these locations been affected?

9) Figure 3: the authors show in Figure 1 VEGF IHC in sections. This stain should be used to demonstrate where the VEGF protein is reduced.

10) Figure 4: how quickly do the CNVs develop in the transgenic models? In both transgenic strains (*Vmd*_*2*_*-Cre* and *iCre-75*) the ‘target tissues’ (i.e., the RPE and photoreceptors) seems to be profoundly affected. We can’t be certain there is not a primary effect (because in these experiments *sFlt-1* and *mFlt-1* are absent). A temporal study showing early vascular changes in the absence of changes in the transgenic target tissues could exclude this.

11) One recurring piece of data is that CNV occurs in the control eyes, although at a lower incidence. It is never addressed why this happens. This should be discussed.

12) For patient samples studied in Figure 1, please present a table summarizing each patient’s age and clinical characteristics.

13) A critical control experiment for recombination efficiency and location is needed for the subretinal injection of Cre plasmid, and the authors have the components for this experiment so it should not be difficult to perform. Indeed, they performed this control for the RPE and photoreceptor-specific lines. The experiment is to inject mTmG mice with subretinal Cre plasmid and see where and at what efficiency Cre-mediated recombination takes place. It would be nice to see this both in cross-section and in flatmounts of isolated retinas and eye-cups with the retina removed (to see the RPE en face). Subretinal injection of DNA has not been previously reported to produce particularly high efficiencies of DNA uptake. In this regard, we wonder why the authors did not use subretinal injection of AAV-cre, a mode of gene delivery that gives high efficiency transduction into RPE cells.

---

## [Author Response]

*1) First, we would like to make a philosophical point. The authors should ensure that their central claim rests on experiments that are objectively reproducible or falsifiable. This field has been beset by data that do not meet that bar. For example, if other scientists fail to reproduce the subretinal injection experiments in which the outcome is CNV volume (e.g., Figure 3), the authors could claim that special skill is required to do the micro-injection correctly. Therefore, the reviewers put more weight on experiments that require no special manipulation, in particular the RPE-specific and photoreceptor-specific* Flt-1 *KOs. Other scientists should be able to breed these lines of mice, perform the histology, and see the neovascularization. If other scientists do not obtain the same result, then that would falsify the central claim of this paper. For this reason, the authors should present a quantification of the neovascularization events per retina for this particular set of experiments. Examples are shown in Figure 4 but one cannot tell whether this represents a common or rare occurrence in these eyes*.

We appreciate the reviewers’ constructive suggestions. In order to put more weight on experiments that require no special microsurgical manipulation, we have added more information in the revised manuscript and included a quantification of neovascular events as follows (indicated with italics at the end): “At 1-3 months of age, nearly all homozygous RPE-specific *Flt-1* knockout mice (*Vmd*_*2*_*-cre*^*+*^*flt-1*^*lox/lox*^) (18/22 eyes, 82%, *P=*1.3E-6), and about half of the heterozygous conditional *Flt-1* knockout mice (*Vmd*_*2*_*-cre*^+^
*flt-1*^*lox/+*^) (17/42 eyes, 40%, *P=*0.009) developed CNV, which progressed over time, compared with 18% (2/22 eyes, 9%) of littermate controls (*Vmd*_*2*_*-cre*^*+*^*flt-1*^*+/+*^). Similarly, at 1-3 months of age, both homozygous (*iCre-75*^*+*^*flt-1*^*lox/lox*^*,* 8/14 eyes, 57%, *P<*0.05) and heterozygous (13/24 eyes, 54%, *P<*0.05) photoreceptor-specific *Flt-1* knockout mice (*iCre-75*^*+*^*flt-1*^*lox/+*^) developed RAP, compared with 10% (1/10 eyes) of littermate controls *(iCre-75*^*+*^*flt-1*^*+/+*^). *FA was performed starting from P21 and we found that neovascular lesions could be detected as early as P28. In CNV or RAP eyes, the lesions appeared in variable numbers and sizes. A typical lesion is shown in Figure 4, and the numbers of leakages in each eye range from 1-4 per eye. Some lesions appeared as scattered little spots (Figure 4—figure supplement 3).*

Furthermore, the other comments on the knockout experiments were addressed in the appropriate context in the revised manuscript.

*2) Regarding RPE vs photoreceptor* sFlt-1 *loss, it is hard to believe that loss of* sFlt-1 *from one cell type would not be compensated for by diffusion of sFLT-1 made by the other cell type*.

While diffusion of sFLT-1 between the RPE and photoreceptors is theoretically possible, the intervening inter photoreceptor matrix could trap sFLT-1, preventing free diffusion. In addition, sFLT-1 secretion by the RPE is polarized basally towards the choroid (Figure 1). Therefore, excessive VEGF secretion from the RPE may overcome the normal anti-angiogenic balance if sFLT-1 is reduced, eliciting choroidal neovascularization. We have added the above to the revised manuscript.

*3) It is not clear how exactly CNVs were quantified. We have the impression OCT was used as the main readout for CNV, which is questionable. Or was fluorescein or ICG data included in the quantification? If yes, how was this done? And how exactly was the OCT data quantified? Looking at the OCTs in Figure 1, we would struggle to quantify changes here. Most importantly, how can CNVs be distinguished in this image modality from fluid build-up due to leakage? Why have the authors not used IHC to visualize the CNVs in whole mounts? This would make the CNV quantification data much more reliable. A similar criticism applies to Figure 3. Why have the authors not used a vessel specific stain to show CNVs? It is impossible to be certain that vascular changes have occurred based on the H&E stains shown. The “V” in CNV stands for “vascularization”, yet the authors do not show any vascular specific stains (apart from Figure 4). This is a major weakness*.

We apologize for not presenting sufficient information. We measured the CNV volumes in vivo by using optical coherence tomography (OCT) images according to established procedure (Schmucker Schaeffel, 2004; Joeres and Tsong et al., 2007; Furino and Ferrara et al., 2009; Ruggeri and Tsechpenakis et al., 2009; Giani and Thanos et al., 2011; Luo and Zhang et al., 2013). This method has been widely used for estimating CNV volumes in vivo, especially in AMD patients and has proved to provide highly reproducible quantitative measurements (Joeres and Tsong et al., 2007; Giani and Thanos et al., 2011; Luo and Zhang et al., 2013). In vivo measurements offer the advantage of following a single animal over time, which cannot be achieved by destructive histologic techniques. We added the above references to the revised manuscript. We compared this method to IHC staining on flatmounts and found good correlation (*r*=0.79, n=19, p=0.0006, data were not shown in this manuscript).

Figure 1 is a histological image of RAP from a human sample, not CNV. We quantified all CNV volumes but not RAP volumes in the experiment. It is hard to confirm the three-dimensional boundary of CNV from FA or ICG leakage but this can readily be done in a stack of OCT slices in vivo.

With respect to vascular staining, we now show in Figure 3—figure supplement 4 that the lesions do contain CNV by isolectin staining. This is additionally corroborated by TEM (Figure 3—figure supplement 4).

*4) Figure 2: the authors should more carefully characterize how* sFlt-1*, the blocking antibody, VEGF, and the anti VEGF antibodies may affect their VEGF and PlGF quantification. For instance, “free VEGF” is mentioned. How can the authors be sure that their ELISA does not measure “non-free” VEGF (bound to* sFlt-1*)? The authors seem to imply that the* sFlt-1 *blocking antibody can displace VEGF from bound* sFlt-1 *(thereby raising VEGF levels). This is a bit surprising, considering the very high affinity* sFlt-1 *has for VEGF. The authors should demonstrate that this is possible for VEGF (but not for PlGF) in vitro*.

*5) Figure 2: how can the authors explain the fact that PlGF is not changed? Similar questions as above arise. Is the anti-*sFlt-1 *blocking antibody not capable of displacing PlGF (but it can do that with VEGF)? Also, can the authors be sure that they are measuring true “free PlGF” (and not PlGF bound to* sFlt-1*)*?

We performed a series of experiments to address the above two comments. First, we determined whether the neutralizing anti-FLT-l antibody affected the measurement of mouse VEGF-A and PlGF-2 (mice only have PlGF-2, not 1 and 3) by ELISA (Figure 2—figure supplement 1). We did not find any significant difference.

Next, to determine whether our ELISA would detect VEGF-A that is bound to FLT-1, we examined the effect of excess recombinant FLT-1 protein on detection of VEGF-A and PlGF-2 by ELISA (Figure 2—figure supplement 2). In this assay, almost all VEGF-A (62.5pg/ml) would be expected to bind FLT-1 (100 ng/ml) based on an assumed Kd=10pM (Free VEGF-A can be estimated to be 1.36pg/ml by Michaelis-Menten kinetics). As shown in Figure 2—figure supplement 2, the ELISA showed less detection of VEGF-A and PLGS after saturation with excess recombinant FLT-1: i.e., it did not detect non-free VEGF-A and non-free PlGF-2. *These data demonstrate that non-free VEGF or non-free PLGF-2 is not being detected at significant levels by our ELISA technique.*

Finally, we determined whether anti-FLT-1 neutralizing antibody released VEGF-A but not PlGF-2 from recombinant FLT-1 (Figure 2—figure supplement 3). After coating ELISA plates with FLT-1 and then incubating with VEGF-A, FLT-1 neutralizing antibody was added with VEGF-A. The binding of VEGF-A to the recombinant FLT-1 was evaluated in the presence and in the absence of anti-FLT-1 antibody. Isotype IgG was used as a control. The same assay was performed with PlGF-2. *The results indicated that FLT-1 neutralizing antibody released VEGF-A but not PlGF-2 from recombinant FLT-1*. Although the exact reason for this is unknown, it is possible that FLT-1 neutralizing antibody may change the FLT-1 conformation, affecting binding of FLT-1 to VEGF-A but not PlGF-2.

We have added the above to the revised manuscript. Furthermore, it should be noted that there is little biologic relevance of effects on PlGF. Figure 2 shows that the PlGF level (0.8 pg/µg of protein extract with a very high error bar) is very low compared to that of VEGF (7 pg/µg of protein extract). Further, over-expression of PlGF induce by AAV injected into subretinal space does not alter at all the CNV volume (Tarallo and Bogdanovich et al., 2012).

*6) Figure 3: the authors claim their shRNA approach does not affect* mFlt-1*. Why have they not shown this in the Western blot in panel A (e.g., instead of β-actin)? This would be much more convincing*.

We performed an experiment to demonstrate that AAV.shRNA.*sFlt-1* specifically affects sFLT-1 but not mFLT-1 by Western blot (see Figure 3—figure supplement 1); this has been included in the Results in the revised manuscript. Furthermore, the specificity of the shRNA used in this construct was demonstrated previously (Ambati and Nozaki et al., 2006).

*7) Figure 3: why does the* sFlt-1 *IHC look so completely different in Figure 3 compared to Figure 1? In Figure 3 the RPE has similar intensity to the IS, whereas in Figure 1 the RPE is much weaker*.

We did observe that the sFLT-1 IHC signal became stronger after subretinal injection with treatment or control (Figure 3) eyes compared to normal eyes without injury (Figure 1). The higher sFLT-1 expression might be due to the inflammatory reaction from injury. We have added the above to the revised manuscript.

*8) Figure 3: how far towards the inner retina is* sFlt-1 *reduced? The authors show in Figure 3 only a very small part of the retina that does not include the IN, ON, and RGC layer. However, Figure 1 suggests that there is significant expression in these locations. Have these locations been affected*?

We observed no changes of sFLT-1 expression in inner retina in the two conditional *Flt*-*1*knockouts. However, sFLT-1 decreased somewhat in the inner retina following subretinal injection withAAV.shRNA.*sFlt-1*. This may be due to the diffusion and leakage of injection along the injection track. We have added the above to the revised manuscript.

*9) Figure 3: the authors show in Figure 1 VEGF IHC in sections. This stain should be used to demonstrate where the VEGF protein is reduced*.

Figure 1 shows the distribution of sFLT-1 and VEGF in normal whole retina, and the distribution of the ratio of sFLT-1/VEGF (Figure 1). A “theoretical” implication is that VEGF is prominent in the inner vascularized retina, the layer with blood vessels and neurons, while *sFlt-1* is prominent in the outer avascular retina. In order to show that the VEGF protein is indeed decreased by subretinal delivery of Cre into *Vegfa*^*loxp/loxp*^ mice, we injected pCre (+10% NeuroPORTER) into the subretinal space of *Vegfa*^*loxp/loxp*^ mice. The results (see Figure 3—figure supplement 1) show selective reduction of VEGF protein in the RPE cells. We added this result in the appropriate context within the revised manuscript. Western blot results confirmed Cre expression as well (Figure 3—figure supplement 5).

*10) Figure 4: how quickly do the CNVs develop in the transgenic models? In both transgenic strains (*Vmd_2_-Cre *and* iCre-75*) the “target tissues” (i.e., the RPE and photoreceptors) seems to be profoundly affected. We can’t be certain there is not a primary effect (because in these experiments* sFlt-1 *and* mFlt-1 *are absent). A temporal study showing early vascular changes in the absence of changes in the transgenic target tissues could exclude this*.

The reviewers raise a good point: pathological disturbances in RPE or photoreceptors due to *Flt-1* knockout may themselves contribute to the development of CNV. It should be noted that there are several inherited retinal degeneration models with changes in RPE/photoreceptors where CNV does not develop – therefore it is not a foregone conclusion that any pathological disturbances in these cells necessarily induces CNV. In the data previously presented, we found that the retina and RPE were normal in places where there was no CNV; however, CNV by its nature is anatomically disruptive to the RPE, which in turn leads to photoreceptor disruption. As *Flt-1* knockdown involves decreased *mFlt-1* as well, it is possible that genomic knockout could induce early profound retinal changes which secondarily induces CNV or RAP.

Hence, as recommended, we conducted a temporal study. We performed immunohistochemistry on sections of VMD-*Creflt*^*loxp/loxp*^ mice at p7, p14, and p21. There were no CNV lesions seen on retinal sections at these time points. However, we found evidence of nascent choroidal endothelial changes abutting Bruch’s membrane at P21 (Figure 4—figure supplement 7). At p21, the earliest feasible time point for angiography (due to time of weaning), fluorescein angiography detected no CNV (Figure 4—figure supplement 8).The earliest time point that we detected CNV angiographically was at P28, when the overlying neural retina appears intact (Figure 4—figure supplement 9). Taken together, these data present early choroidal vascular changes without changes in the overlying neural retinal changes.

*11) One recurring piece of data is that CNV occurs in the control eyes, although at a lower incidence. It is never addressed why this happens. This should be discussed*.

Rare CNV occurrences did occur in the littermate controls (Tables 2 and 3). These results may be due to Cre “leakiness” or toxicity which has been reported in several transgenic models using the Cre-lox system (14; 50).

*12) For patient samples studied in Figure 1, please present a table summarizing each patient’s age and clinical characteristics*.

The demographic characteristics of the patient samples are shown in Table 1 in the revised manuscript.

*13) A critical control experiment for recombination efficiency and location is needed for the subretinal injection of Cre plasmid, and the authors have the components for this experiment so it should not be difficult to perform. Indeed, they performed this control for the RPE and photoreceptor-specific lines. The experiment is to inject mTmG mice with subretinal Cre plasmid and see where and at what efficiency Cre-mediated recombination takes place. It would be nice to see this both in cross-section and in flatmounts of isolated retinas and eye-cups with the retina removed (to see the RPE en face). Subretinal injection of DNA has not been previously reported to produce particularly high efficiencies of DNA uptake. In this regard, we wonder why the authors did not use subretinal injection of AAV-cre, a mode of gene delivery that gives high efficiency transduction into RPE cells*.

The use of 10% NeuroPORTER Transfection Reagent (Genlantis, San Diego, CA, USA) as a plasmid transfection enhancer has been reported to achieve high DNA uptake in the RPE (Kachi and Oshima et al., 2005). In addition, we have previously published on the successful use of this technique (Kaneko et al. Nature 2011; Tarallo et al. Cell 2012). The experiment for comment #9 demonstrated the efficacy of this approach.

References

Schmucker, C. and F. Schaeffel 2004 *A paraxial schematic eye model for the growing C57BL/6 mouse*. 1857-1867 doi: 10.1016/j.visres.2004.03.011

Joeres, S., J. W. Tsong, P. G. Updike, A. T. Collins, L. Dustin, A. C. Walsh, et al. 2007 *Reproducibility of quantitative optical coherence tomography subanalysis in neovascular age-related macular degeneration*. 4300-4307 doi: 10.1167/iovs.07-0179

Furino, C., A. Ferrara, N. Cardascia, G. Besozzi, G. Alessio, L. Sborgia, et al. 2009 *Combined cataract extraction and intravitreal bevacizumab in eyes with choroidal neovascularization resulting from age-related macular degeneration*. 1518-1522 doi: 10.1016/j.jcrs.2009.04.032

Ruggeri, M., G. Tsechpenakis, S. Jiao, M. E. Jockovich, C. Cebulla, E. Hernandez, et al. 2009 *Retinal tumor imaging and volume quantification in mouse model using spectral-domain optical coherence tomography*. 4074-4083

Giani, A., A. Thanos, M. I. Roh, E. Connolly, G. Trichonas, I. Kim, et al. 2011 *In vivo evaluation of laser-induced choroidal neovascularization using spectral-domain optical coherence tomography*. 3880-3887 doi: 10.1167/iovs.10-6266

Luo, L., X. Zhang, Y. Hirano, P. Tyagi, P. Barabas, H. Uehara, et al. 2013 *Targeted Intraceptor Nanoparticle Therapy Reduces Angiogenesis and Fibrosis in Primate and Murine Macular Degeneration*. doi: 10.1021/nn305958y

Tarallo, V., S. Bogdanovich, Y. Hirano, L. Tudisco, L. Zentilin, M. Giacca, et al. 2012 *Inhibition of choroidal and corneal pathologic neovascularization by Plgf1-de gene transfer*. 7989-7996 doi: 10.1167/iovs.12-10658

Ambati, B. K., M. Nozaki, N. Singh, A. Takeda, P. D. Jani, T. Suthar, et al. 2006 *Corneal avascularity is due to soluble VEGF receptor-1*. 993-997 doi: 10.1038/nature05249

Editorial 2007 *Toxic alert*. 378 doi: 10.1038/449378a

Schmidt-Supprian, M. and K. Rajewsky 2007 *Vagaries of conditional gene targeting*. 665-668 doi: 10.1038/ni0707-665

Kachi, S., Y. Oshima, N. Esumi, M. Kachi, B. Rogers, D. J. Zack, et al. 2005 *Nonviral ocular gene transfer*. 843-851 doi: 10.1038/sj.gt.3302475

Luo, L., X. Zhang, Y. Hirano, P. Tyagi, P. Barabas, H. Uehara, et al. 2013 *Targeted Intraceptor Nanoparticle Therapy reduces Angiogenesis and Fibrosis in Primate & Murine Macular Degeneration*. doi: 10.1021/nn305958y